# THE ENSEMBLE INVERSE PROBLEM: APPLICATIONS AND METHODS

## ABSTRACT

We introduce a new multivariate statistical problem that we refer to as the Ensemble Inverse Problem (EIP). The aim of EIP is to invert for an ensemble that is distributed according to the pushforward of a prior under a forward process. In high energy physics (HEP), this is related to a widely known problem called unfolding, which aims to reconstruct the true physics distribution of quantities, such as momentum and angle, from measurements that are distorted by detector effects. In recent applications, the EIP also arises in inverse imaging with unknown priors. We propose *non-iterative inference-time methods* that construct posterior samplers based on a new class of conditional generative models, which we call ensemble inverse generative models. For the posterior modeling, these models additionally use the ensemble information contained in the observation set on top of single measurements. Unlike existing methods, our proposed methods avoid explicit and iterative use of the forward operator at inference time via training across several sets of truth-observation pairs that are consistent with the same forward operator, but originate from a wide range of priors. We demonstrate that this training procedure implicitly encodes the likelihood model. The use of ensemble information helps posterior inference and enables generalization to unseen priors. We benchmark the proposed method on several synthetic and real datasets in HEP and inverse imaging.

## 1   INTRODUCTION

Let $x \in \mathbb{R}^d$ be a random variable with a prior distribution $p(x)$. We make an observation $y$ about the truth $x$ via a *forward model*:

$$y = F(x) + n(x), \qquad \text{(Fwd-Model)}$$

where $F$ is a forward (measurement) operator and $n(x)$ represents an additive noise, which can in general depend on $x$. Within this setup, we consider the following problem that we refer to as the Ensemble Inverse Problem (EIP). We are given a dataset $\mathcal{D} = \{\mathcal{D}_1, \cdots, \mathcal{D}_M\}$ consisting of multiple truth-observation pairs arising from sampling observations via equation Fwd-Model from $M$ prior distributions $p_m, m \in [1 : M]$.

$$\mathcal{D}_m = \{(x^{m,j}, y^{m,j})\}_{j=1}^{N_m} \overset{i.i.d.}{\sim} p_m(x)p(y|x), \qquad (1)$$

where $(x^{m,j}, y^{m,j})$ denotes the $j$-th truth-observation pair in $\mathcal{D}_m$, and the size of $\mathcal{D}_m$ is $N_m$. The pair $(x^{m,j}, y^{m,j})$ is independently and identically distributed (i.i.d.) according to the joint distribution $p_m(x)p(y|x)$, and the conditional distribution $p(y|x)$ is determined via equation Fwd-Model and is the same for all datasets $\{\mathcal{D}_1, \cdots, \mathcal{D}_M\}$. We assume that we only have access to $\mathcal{D}$ and no direct knowledge about equation Fwd-Model.

**Problem statement (EIP-I for the prior):**   Given training data $\mathcal{D}$, and given a new set of observations $\mathcal{Y} = \{y^1, \cdots, y^N\}$ obtained from an unknown prior $p(x)$ and the same (as $\mathcal{D}$, but unknown) forward model, generate samples $x^1, \cdots, x^{N'}|\mathcal{Y}$ such that for a given $\lambda > 0$,

$$\rho(\hat{p}(x|\mathcal{Y}), p(x)) < \lambda,$$

where $\hat{p}(x|\mathcal{Y}) = \lim_{N' \to \infty} \frac{1}{N'} \sum_{n=1}^{N'} \delta_{x^n|\mathcal{Y}}$ is the limiting empirical measure corresponding to the generated samples. $\rho(\cdot, \cdot)$ denotes a discrepancy measure between distributions, such as the

Kullback-Liebler divergence, total Variation Cover & Thomas (2006), or Wasserstein distance Villani (2009) ,and the Dirac delta function $\delta_{x^n}$ denotes the probability density of a distribution concentrated at the $n$-th generated sample $x^n$. In other words, the aim of EIP-I is to generate samples whose distribution comes close to the prior distribution that lead to the observations. For practical utility that will become clear in the exposition later, we restrict the EIP-I problem further to learn to generate samples via posterior sampling, given observations from a prior.

**Problem statement (EIP-II for the posterior):** Given training data $\mathcal{D}$, and given a new set of *i.i.d.* observations $\mathcal{Y} = \{y^1, \cdots, y^N\}$ obtained from an unknown prior $p(x)$ and the same (as $\mathcal{D}$) but unknown forward model, for any given $y$, generate conditional samples $x^1, \cdots, x^{N'}|y, \mathcal{Y}$ such that for a given $\lambda > 0$,

$$\rho(\hat{p}(x|y, \mathcal{Y}), p(x|y)) < \lambda,$$

where $\hat{p}(x|y, \mathcal{Y}) = \lim_{N' \to \infty} \frac{1}{N'} \sum_{n=1}^{N'} \delta_{x^n|y, \mathcal{Y}}, p(x|y) = \frac{p(x)p(y|x)}{p(y)}$.

It is evident that the integration of the solution to EIP-II yields a good approximation of the solution to EIP-I. However, the integration of posteriors that are not the solution to EIP-II can still be the solution to EIP-I. We refer the readers to the Gaussian example in Sec. 3 in Butter et al. (2025) and our example in Fig. 1.

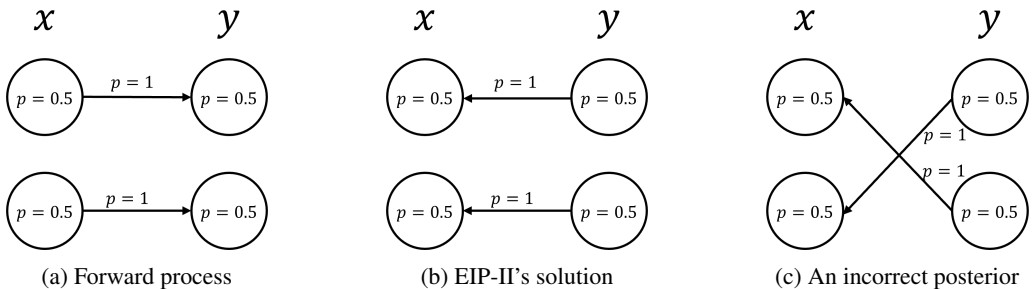

(a) Forward process        (b) EIP-II's solution        (c) An incorrect posterior

Figure 1: Consider a forward process in Fig. 1a, Fig. 1b shows EIP-II's solution, with its integration corresponding to EIP-I's solution. Fig. 1c shows an incorrect posterior; however, the integration of this incorrect posterior can lead to the correct prior.

Owing to the success of the generative models in modeling complex distributions with provable theoretical guarantees Ho et al. (2020); Chen et al. (2023); Albergo & Vanden-Eijnden (2023), in this paper, we aim to solve EIP-II by modeling the posterior via generative models.

**Where does EIP arise?** An important application of EIP arises in the high-energy physics (HEP) domain, where one *unfolds* to remove detector effects D'Agostini (2010); Andreassen et al. (2020). A point of distinction in our problem statement and the traditional unfolding setup is that EIP-I & II do not make explicit use of the forward operator at inference time. The primary reason to deviate from such a setting is that typically forward models are computationally expensive to simulate. So EIP-I & II provide for an avenue where this model is implicit in the dataset $\mathcal{D}$. In the context of unfolding, EIP-II setting has recently been considered directly in Pazos et al. (2025) using conditional generative models.

Another domain where EIP-II arises naturally is the inverse imaging problem setting, where one wants to recover a corrupted image with an unknown prior Daras et al. (2023); Hu et al. (2024). A set of recent works has considered EIP-like problems arising in contexts of Large-Language Models and the In-Context Learning Geshkovski et al. (2024); Teh et al. (2025); Adu & Gharesifard (2024). In Geshkovski et al. (2024); Adu & Gharesifard (2024) the main problem is to understand if given pairs of measures whether there exists a transformer architecture that can map a given input to its corresponding output, thus learning a measure to measure map. The setting of Teh et al. (2025) also comes close to EIP. Teh et al. (2025) proposes to use a transformer to infer the hidden parameters in a Poisson forward process, provided with a set of observations.

We now summarize related work in terms of the methods that have been proposed in the literature and which can potentially be used to address the EIP problem.

| | Method | Requirements | Objective | Iterative | Tuneable Regularization | Designed to recover unseen priors |
|---|---|---|---|---|---|---|
| Non-ML | IBU | equation Fwd-Model | $p(x)$ | Yes | Yes | Yes |
| | SVD Unfolding | equation Fwd-Model | $p(x)$ | Partial | Yes | Yes |
| | Measure decomposition | equation Fwd-Model | $p(x\|y)$ | Yes | Yes | Yes |
| Thoretical | Measure-to-measure interpolation | $\mathcal{D}$ | $p(x)$ | No | No | No |
| ML-based | OmniFold | $\mathcal{D}$ or equation Fwd-Model | $p(x)$ | Yes | Yes | Yes |
| | GANs | equation Fwd-Model | $p(x)$ | No | No | Yes |
| | DPnP | $\{x^j\}_{j=1}^N$ and equation Fwd-Model | $p(x\|y)$ | No | No | No |
| | Ambient diffusion | $\{y^j\}_{j=1}^N$ and equation Fwd-Model | $p(x\|y)$ | No | No | No |
| | cINN | equation Fwd-Model | $p(x\|y)$ | No | No | No |
| | SBUnfold | $\mathcal{D}$ | $p(x\|y)$ | No | No | Yes |
| | DDRM | Pretrained model and equation Fwd-Model | $p(x\|y)$ | No | No | No |
| | GDDPM | $\mathcal{D}$ | $p(x\|y)$ | No | No | Yes |
| | Ours | $\mathcal{D}$ | $p(x\|y)$ | No | No | Yes |

Table 1: Comparison of methods for solving EIP-I (objective: $p(x)$) & EIP-II (objective: $p(x|y)$) and their key characteristics. Iterative Bayesian unfolding (IBU) appears in D'Agostini (2010). Singular value decomposition (SVD) Unfolding appears in Höcker & Kartvelishvili (1996). Measure decomposition method for posterior sampling appears in Montanari & Wu (2025). Measure-to-measure interpolation approaches appear in Geshkovski et al. (2024); Adu & Gharesifard (2024). OmniFold appears in Andreassen et al. (2020). Generative adversarial networks (GANs) for inverse problems appear in Bellagente et al. (2020b); Datta et al. (2018). Diffusion plug-and-play (DPnP) method appears in Xu & Chi (2025). Ambient diffusion appears in Daras et al. (2023). Conditional invertible neural networks (cINN) approaches appear in Backes et al. (2024); Heimel et al. (2024); Bellagente et al. (2020a). SBUnfold appears in Diefenbacher et al. (2023). Denoising diffusion restoration model (DDRM) appears in Kawar et al. (2022). Generalizable conditional denoising diffusion probabilistic model (GDDPM) appears in Pazos et al. (2025).

## 1.1 RELATED WORKS

Table 1 provides a summary of key features among non-ML, theoretical, and ML-based methods for solving the EIP and / or classical inverse problem.

1. **Non-ML methods:** Traditional methods designed for unfolding reconstruct the prior via iterative probabilistic updates IBU D'Agostini (2010) and suppression of contributions with small singular values Höcker & Kartvelishvili (1996). Common features of them include relying on explicit modeling of the forward process and requiring the data to be binned. In a more general setting, Montanari & Wu (2025) proposes an iterative posterior measure decomposition method that enables efficient sampling for sparse Bayesian inverse problems.

2. **Theoretical methods:** Geshkovski et al. (2024); Adu & Gharesifard (2024) provide mathematical frameworks for understanding transformers as measure-to-measure maps and prove that a single transformer can approximate the transport maps and velocity fields between multiple distribution pairs. The depth and complexity of the transformer depend on the structure and the number of pairs. However, the problem of generalization to unseen measures was not considered, and no algorithm was proposed for solving the EIP. Teh et al. (2025) proves that transformers can approximate classical empirical Bayes estimators and proposes a training algorithm. Nevertheless, this method is limited to the one-dimensional Poisson–EB setting.

3. **ML-based methods:** Omnifold Andreassen et al. (2020) is a representative iterative re-weighting method for unfolding that shapes a given prior to the target prior. Generative methods have also become successful tools for addressing inverse problems, leading to a surge of approaches, including GANs Bellagente et al. (2020b); Datta et al. (2018), DPnP Xu & Chi (2025), ambient diffusion, and SBUfold Diefenbacher et al. (2023). In particular, GDDPM Pazos et al. (2025) aims to solve EIP-II via posterior modeling and sampling. Built based on conditional DDPM (cDDPM), GDDPM additionally utilizes moment information of observations to ensure generalization ability across different physics processes. With the objective of avoiding computationally costly iterative inference, bypassing the difficulty of obtaining the forward operator, and effectively incorporating distributional information embedded in observations, this work provides a framework for solving EIP-II via generative models.

## 1.2 CONTRIBUTIONS

We list the contributions of this work as follows,

1. This work proposes a novel non-iterative framework for solving EIP-II, called ensemble inverse generative models, which models the posterior sampling process and is conditioned on both measurements and observation sets.

2. With the ensemble information extracted via a permutation invariant structure from the observation set, the proposed method demonstrates a superior posterior inference ability and a strong generalization ability to unseen priors.

3. Under several synthetic settings and real applications, including HEP unfolding and image inversion tasks, we demonstrate that the proposed methods outperform baselines without relying on explicit knowledge about the priors and the forward model.

## 2 METHOD

We address EIP-II via a non-iterative posterior sampling method. Specifically, generative models that are conditioned on not only the single measurement $y$ but also the observation set $\mathcal{Y}$, are utilized to model the posterior and serve as a posterior sampler. With the aid of ensemble information extracted from the observation set $\mathcal{Y}$, the proposed method is shown to have a strong inductive bias to unseen priors. To state the methods, we refer the readers to two successful generative models, viz., generative models, Denoising Diffusion Probabilistic Models (DDPM) Ho et al. (2020) and Flow Matching (FM) Lipman et al. (2023) for backgrounds, and we provide more details for the conditional version of them in Sec. A.

### 2.1 ENSEMBLE INVERSE GENERATIVE MODELS FOR EIP-II

Our main idea behind addressing EIP-II is that the observation set $\mathcal{Y}$, in which all observations yield from a single prior distribution $p(x)$, contains information about $p(x)$. This prior information is not directly available, but can contribute towards a valid posterior inference for any given $y$ yielding from $p(x)$. Inspired by Teh et al. (2025); Pazos et al. (2025) and with the objective of utilizing the ensemble information contained in $\mathcal{Y}$, our recovery model is conditioned on not only the measurement $y$ but also the observation set $\mathcal{Y}$. The size of $\mathcal{Y}$ should generally be large in order to reflect the underlying ensemble information. However, in conditional generative modeling, directly conditioning on a large input set can be computationally inefficient and statistically unstable, as the model must process high-dimensional and unordered data. To address this, one can first encode the set using a *permutation invariant* structure, such as using the moment function as in Pazos et al. (2025). For a more versatile and adaptive representation, we propose to extract the ensemble information via $\phi_w : \mathbb{R}^{N \times d} \to \mathbb{R}^k$, a permutation invariant neural network (NN) parameterized with $w$, that maps an observation set $\mathcal{Y}$ containing $N$ $d$-dimensional samples into a $k$-dimensional representation that reflects the ensemble information. Formally, let $S_N$ denote the set of all permutation of indices $\{1, 2, \cdots, N\}$. $\phi_w$ should satisfy

$$\forall s \in S_N, \quad \phi_w(s\mathcal{Y}) = \phi_w(\mathcal{Y}), \quad \mathcal{Y} = \{y^1, \cdots, y^N\}. \tag{2}$$

This allows $\phi_w$ to process $\mathcal{Y}$ as a set, focusing on the group feature and ignoring the order information. Optional choices for implementing $\phi_w$ include deep set Zaheer et al. (2017) and set transformer Lee et al. (2019).

Based on this insight, we propose an algorithm for solving EIP-II, named ensemble inverse denoising diffusion probabilistic model (EI-DDPM) / ensemble inverse flow matching (EI-FM), as presented in Alg. 1 and Alg. 2. EI-DDPM / EI-FM is based on conditional-DDPM / conditional-FM frameworks, wherein an NN denoted by $\varepsilon_\theta$, parameterized by $\theta$ is employed to predict the noise / velocity field at each step. In addition to the intermediate states $x_t$ and time information $t$, $\varepsilon_\theta$ accepts single measurements $y$, as well as the ensemble information $\phi_w(\mathcal{Y})$ as inputs in order to model the posterior $p(x|y, \mathcal{Y})$ in EIP-II. Although the dimension of the ensemble information $k$ is determined by the user, we emphasize here that $k$ should be generally set close to $d$ for a balanced input of $y \in \mathbb{R}^d$ and $\phi_w(\mathcal{Y}) \in \mathbb{R}^k$ into the generative models. The incorporation of $\phi_w(\mathcal{Y})$ facilitates the posterior inference for measurements $y$. Provided with truth-observation pairs resulting from sufficiently diverse priors, $\varepsilon_\theta$ and $\phi_w$ combined is able to generalize for posteriors induced by previously unseen priors. We numerically illustrate these features in Sec. 3.

The stability of the learned representation of ensemble information $\phi_w(\mathcal{Y})$ depends on an extra hyperparameter $N$ – the number of samples in $\mathcal{Y}$. First, $N$ should be large enough for $\mathcal{Y}$ to have

the capability to represent the distributional information of $p(y)$, thus being able to contain valid ensemble information. Second, considering that $N$ is fixed during the training stage in Alg. 1, the input observation set size for Alg. 2 of inference should remain $N$ for robustness. Therefore, it is important to discuss cases in which the available observation set size $N' \neq N$, at inference time. For the case $N' > N$, subsets of size $N$ can be picked repeatedly to perform Alg. 2 until the union of the subsets fully covers the target observation set. For the case $N' < N$, one can randomly duplicate $N - N'$ samples so that the set size is expanded to $N$. For target sets with $N' \ll N$, Alg. 2 with duplication strategy may perform in a bad way since a set with too many duplicates will reflect highly incorrect ensemble information. The effects of $N$ and $N'$ are further discussed and numerically investigated in Sec. B.2.

---

**Algorithm 1** EI-DDPM's and EI-FM's Training algorithm

---

**Input:**
  $\varepsilon_\theta, \phi_w, N, \mathcal{D} = \{\mathcal{D}_1, \cdots, \mathcal{D}_M\}$, EI-DDPM's schedule parameters $\{\beta_t, \alpha_t, \bar{\alpha}_t, T\}$, learning rate $\eta$
**Output:** Trained $\varepsilon_\theta, \phi_w$

**repeat**
    Choose $m \sim \text{Uniform}(\{1, \cdots, M\})$
    Draw a $N$ pairs subset $\{(x^{m,j}, y^{m,j})\}_{j=1}^N$ from $\mathcal{D}_m, \mathcal{Y} \leftarrow \{y^{m,j}\}_{j=1}^N$
    **for** each $(x, y)$ pair in the subset **do**
        $\mathcal{L}(\theta, w) \leftarrow 0$
        **if** using EI-DDPM **then**
            $t \sim \text{Uniform}(\{1, \cdots, T\}), \xi \sim \mathcal{N}(\mathbf{0}, \mathbf{I})$
            $\mathcal{L}(\theta, w) \leftarrow \mathcal{L}(\theta, w) + \left\| \varepsilon_\theta \left( \sqrt{\bar{\alpha}_t} x + \sqrt{1 - \bar{\alpha}_t} \xi, t, y, \phi_w(\mathcal{Y}) \right) - \xi \right\|_2^2$
        **else if** using EI-FM **then**
            $t \sim \mathcal{U}[0, 1], \xi \sim \mathcal{N}(\mathbf{0}, \mathbf{I})$
            $\mathcal{L}(\theta, w) \leftarrow \mathcal{L}(\theta, w) + \left\| \varepsilon_\theta(tx + (1 - t)\xi, t, y, \phi_w(\mathcal{Y})) - (x - \xi) \right\|_2^2$
        **end if**
    **end for**
    $(\theta, w) \leftarrow (\theta, w) - \eta \nabla \mathcal{L}(\theta, w)$
**until** converged
**Return** $\varepsilon_\theta, \phi_w$

---

## 3   EXPERIMENTS

### 3.1   BASELINES FOR COMPARISON

**Conditional DDPM (cDDPM) and conditional FM (cFM):**   cDDPM and cFM model the posterior $p(x|y)$ with the conditional variable incorporating only a single measurement. No ensemble information is included.

**GDDPM Pazos et al. (2025):**   GDDPM is built upon cDDPM and it incorporates additional moment information computed from $\mathcal{Y}$.

**Omnifold Andreassen et al. (2020):**   Omnifold is a reweighting-based unfolding method that reweighs a given initial distribution towards the prior. The initial distribution is a critical factor in recovery performance. Since in the EIP setup, we are provided with $\{(x^{m,j}, y^{m,j})\}_{j=1}^{N_m}, m = 1, \cdots, M$, we consider two ways of selecting the initial distribution to invert for a set of observations $\mathcal{Y}'$. a) **Omnifold-best:** Picking $m^\star$, such that $\{y^{m^\star, j}\}_{j=1}^{N_{m^\star}}$ has the minimum sliced Wasserstein distance (SWD)[1] Bonneel et al. (2014) from $\mathcal{Y}'$, and $\{x^{m^\star, j}\}_{j=1}^{N_{m^\star}}$ serves as the initial distribution; and b) **Omnifold-combine:** Using the mixture of all available priors $\{x^{m,j}\}_{j=1}^{N_m}, m = 1, \cdots, M$ as the initial distribution.

---

[1]SWD measures the similarity between two distributions, with smaller values indicating greater similarity.

---

**Algorithm 2** EI-DDPM's and EI-FM's sampling algorithm

---

**Input:** $\varepsilon_\theta$, $\phi_w$, $\mathcal{Y} = \{y^j\}_{j=1}^N$, EI-DDPM's schedule parameters $\{\alpha_t, \bar{\alpha}_t, \sigma_t, T\}$, EI-FM's discretization interval $\Delta t$
**Output:** $\{\hat{x}^j\}_{j=1}^N$

$z = \phi_w(\mathcal{Y})$
**for** $j = 1, 2, \cdots, N$ **do**
    **if** using EI-DDPM **then**
        $x_T \leftarrow \mathcal{N}(\mathbf{0}, \boldsymbol{I})$
        **for** $t = T \cdots, 1$ **do**
            $\xi \leftarrow \mathcal{N}(\mathbf{0}, \boldsymbol{I})$ if $t > 1$, else $\xi \leftarrow 0$
            $x_{t-1} \leftarrow \frac{1}{\sqrt{\alpha_t}} \left( x_t - \frac{1-\alpha_t}{1-\bar{\alpha}_t} \varepsilon_\theta(x_t, t, y^j, z) \right) + \sigma_t \xi$
        **end for**
        $\hat{x}^j \leftarrow x_0$
    **else if** using EI-FM **then**
        $x_0 \leftarrow \mathcal{N}(\mathbf{0}, \boldsymbol{I}), t \leftarrow 0$
        **repeat**
            $t \leftarrow t + \Delta t$
            $x_t \leftarrow x_{t-\Delta t} + \varepsilon_\theta(x_{t-\Delta t}, t, y^j, z)\Delta t$
        **until** $t = 1$
        $\hat{x}^j \leftarrow x_1$
    **end if**
    **end for**
**Return** $\{\hat{x}_j\}_{j=1}^N$

---

**SBUnfold Diefenbacher et al. (2023):** SBUnfold leverages Schrodinger Bridges with diffusion models to map measurements to their truth.

**Sourcerer Vetter et al. (2024):** Sourcerer is a sample-based method for inverse problems that jointly maximizes entropy and minimizes sample-based distance, e.g., SWD, between simulations and data. It requires an available differentiable forward operator or a differentiable surrogate of it. In our cases, an available differentiable forward operator is not directly accessible; however, a surrogate can be trained based on the truth-observation pairs.

The NN structures for cDDPM, cFM, SBUnfold, and $\varepsilon_\theta$ used in EI-DDPM / EI-FM are kept the same (with input dimensions adjusted to match their respective inputs) for a fair comparison. We use the set transformer Lee et al. (2019) structure for the implementation of $\phi_w$.

## 3.2 2-D GAUSSIAN EIP

We first present a toy example of inverting for a perturbed 2-D Gaussian distribution to demonstrate the effectiveness of the proposed method. The prior is a bivariate Gaussian distribution with mean $[0, 0]^\top$ and covariance matrix $\begin{bmatrix} 1 & \gamma \\ \gamma & 1 \end{bmatrix}$, where $\gamma \in [-1, 1]$ represents the the correlation coefficient between the two dimensions. Let $x = [x_1, x_2]^\top \in \mathbb{R}^2$ denote a sample from the prior. The prior is given as

$$x|\gamma \sim \mathcal{N}\left(\begin{bmatrix} 0 \\ 0 \end{bmatrix}, \begin{bmatrix} 1 & \gamma \\ \gamma & 1 \end{bmatrix}\right). \tag{3}$$

In this EIP, we consider that $x$ undergoes a linear transformation by a matrix $A \in \mathbb{R}^{2\times 2}$, and is perturbed by an additive noise term $n(x) \in \mathbb{R}^2$. The observed signal $y \in \mathbb{R}^2$ is given by

$$y = Ax + n(x), \quad A = \begin{bmatrix} 1 & 0.5 \\ 0.5 & 2 \end{bmatrix}, \quad n(x) \sim \mathcal{N}\left(\begin{bmatrix} 0.2x_1 \\ 0.2x_2 \end{bmatrix}, \begin{bmatrix} 0.25\|x\|_2^2 & 0 \\ 0 & 0.25\|x\|_2^2 \end{bmatrix}\right). \tag{4}$$

The objective is to recover the prior given its observation set $\mathcal{Y}$ corresponding to an unknown $\gamma$.

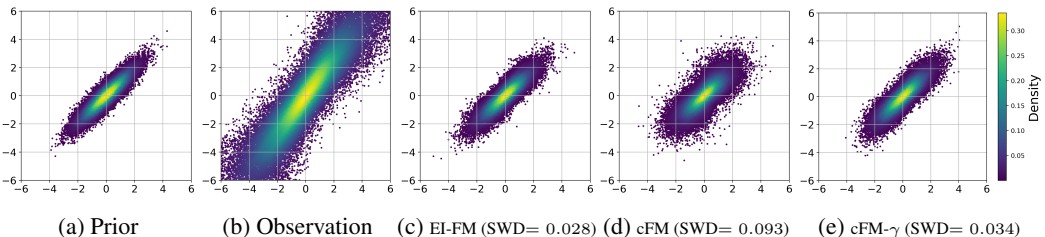

| (a) Prior | (b) Observation | (c) EI-FM (SWD= 0.028) | (d) cFM (SWD= 0.093) | (e) cFM-$\gamma$ (SWD= 0.034) |

Figure 2: Visualization of 40000 samples in the prior ($\gamma = 0.9$) and recovered distributions via various methods.

During the training stage, truth-observation pairs resulting from priors with $\gamma \in [-0.75, -0.25] \cup [0.25, 0.75]$ are provided. In the inference time, we evaluate the recovery performance for priors with $\gamma \in [-1, 1]$ perturbed by equation 4. We mainly focus on FM-based models for comparison to avoid overcrowded results. Observation set size $N = 4000$ and ensemble information dimension $k = 3$ are set for EI-FM, i.e., $\phi_w : \mathbb{R}^{4000 \times 2} \to \mathbb{R}^3$. Besides the mentioned baselines, we also evaluate cFM-$\gamma$, which is built based on cFM, but additionally conditioned on the latent information $\gamma$. cFM-$\gamma$ assumes direct knowledge of the priors.

Fig. 2 visualizes the distribution of the prior with $\gamma = 0.9$ and the recovered distributions by 3 representative methods to illustrate EI-FM's generalization ability for recovering priors in the same parameter family as in the training set. The true prior with $\gamma = 0.9$ is a "thin" distribution, which is unseen during the training time. cFM's recovery is much "wider" than the prior since it performs an element-wise generation without considering the ensemble information. EI-FM, which incorporates the ensemble information from observed samples, can achieve similar performance to cFM-$\gamma$ with direct prior knowledge, illustrating its capability to generalize to unseen distributions.

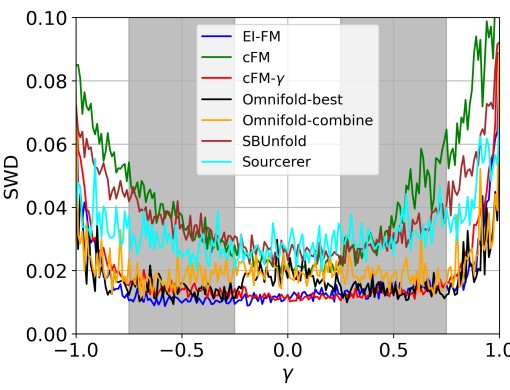

Figure 3: Average sample-wise SWD($\downarrow$) between the truth and the recovery vs. $\gamma$, evaluated over 40000 samples. Grey areas denote the priors contained in the training data.

In Fig. 3, we compare the SWD between the prior and the recovered distributions w.r.t. 40000 samples for each $\gamma$ in $\{-1, -0.99, \cdots, 0.99, 1\}$. EI-FM displays superior recovery performance among all compared methods and behaves close to cFM-$\gamma$, for $\gamma \in [-1, 1]$. Omnifold-best's initial distribution is exact the priors for $\gamma \in [-0.75, -0.25] \cup [0.25, 0.75]$, leading to low SWD. However, Omnifold displays weaker generalization ability than EI-FM for $\gamma \in [-0.25, 0.25]$. Therefore, we can conclude that the EI-FM is able to effectively utilize the ensemble information of observations to help infer the posterior and generalize to unseen distributions with performance comparable to models directly provided with prior information.

### 3.3 PARTICLE PHYSICS DATA UNFOLDING

In this section, we evaluate the proposed methods on simulated particle physics data. The data consists of quantum chromodynamics (QCD) jets, which are collimated sprays of particles produced when partons (the constituent particles within protons) fragment in high-energy collisions. These datasets are generated using the PYTHIA 8.3 event generator Bierlich et al. (2022) for various physics processes such as $t\bar{t}$, $W$+jets, $Z$+jets, dijet, and leptoquark processes. The jet kinematics include transverse momentum ($p_T$), pseudorapidity ($\eta$), azimuthal angle ($\phi$), and 4-momentum components ($E, p_x, p_y, p_z$). These jets are presented at 2 stages: the truth-level ($x$) representation is constructed from the direct output of the Monte Carlo event generator, while the detector-level ($y$) is the representation after the jets pass through the detector simulation. The training data consists

of pairs of truth-level and detector-level jet vector pairs. The truth-level jet vectors come from 18 different physics processes, including various parton distribution functions and parton shower models, and the detector effects are identical across all truth-level jet vectors. We refer readers to Pazos et al. (2025) for more details on this dataset. During inference time, we compare the distribution similarity between the recovered data from 4 unseen physics processes and their truth-level data.

GDDPM Pazos et al. (2025) proposes to incorporate the first 6 moment information of the $p_T$ to help unfolding. However, this implicitly assumes that $p_T$ contains the complete distributional information of the 7-component vector. Therefore, we also consider a more general variant, referred to as GDDPM-v, in which this assumption is not made and moments of all 7 components are taken as the conditional information. The Wasserstein-1 distance (WD) Villani (2009) for each jet kinematics between the true distributions and the recovered distributions is selected as the metric for measuring the distribution similarity following Pazos et al. (2025).

$N = 2000$ and $\phi_w : \mathbb{R}^{2000 \times 7} \to \mathbb{R}^6$ are fixed in both EI-DDPM and EI-FM in this particle physics unfolding task. Fig. 4 showcases the EI-FM's reconstruction of $p_T, E$ and $p_x$ distributions from a $t\bar{t}$ process. The detector effects cause a great difference between the truth and the detector-level distributions. EI-FM is able to recover distributions with small WDs to the truth. Table 2 shows the recovery performances of $p_T, E$ and $p_x$ for 4 unseen physics processes. The proposed methods display superior performances across all 4 unseen physics processes, illustrating the effectiveness of the proposed methods in utilizing latent ensemble information for unfolding without knowledge of the priors. It is worth mentioning that GDDPM outperforms GDDPM-v, suggesting that redundant moment information in GDDPM-v impairs recovery. Nevertheless, our proposed methods achieve comparable or superior performance to GDDPM, indicating that $\phi_w(\cdot)$ can automatically extract the core ensemble information from $\mathcal{Y}$ and eliminate redundant information.

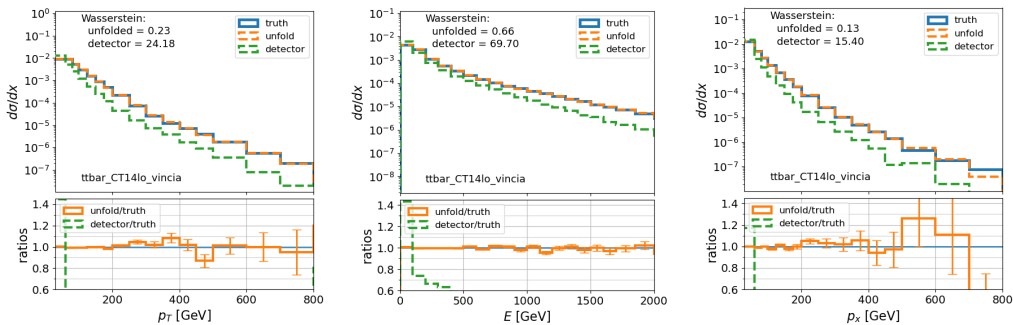

Figure 4: Unfolding results of jet kinematics from a $t\bar{t}$ process (modeled with the CT14lo PDF and Vincia parton showers) from the data-driven detector smearing using EI-FM.

| WD ($\downarrow$) | Name | Detector | EI-DDPM | EI-FM | cDDPM | cFM | GDDPM-v | GDDPM | Omnifold-best | Omnifold-combine | SBUnfold | Sourcerer |
|---|---|---|---|---|---|---|---|---|---|---|---|---|
| $p_T$ | Leptoquark | 31.85 | 0.44 | 0.44 | 1.08 | 2.65 | 0.73 | 0.44 | **0.19** | 0.82 | 18.10 | 7.96 |
| | $t\bar{t}$ (CT14lo, Vincia) | 24.18 | 0.44 | **0.23** | 1.01 | 3.36 | 1.36 | 0.55 | 0.60 | 0.52 | 1.88 | 15.14 |
| | $W$+jets (CT14lo) | 18.60 | 0.60 | 0.44 | 2.41 | 4.14 | 0.53 | 0.48 | **0.11** | 0.37 | 21.07 | 25.96 |
| | $Z$+jets (CTEQ6L1) | 15.81 | 0.51 | **0.45** | 2.55 | 5.55 | 2.25 | 1.98 | 0.48 | 0.64 | 25.18 | 19.59 |
| $E$ | Leptoquark | 83.87 | **0.46** | 0.76 | 4.70 | 2.66 | 1.47 | 0.63 | 1.08 | 0.47 | 13.99 | 57.42 |
| | $t\bar{t}$ (CT14lo, Vincia) | 69.70 | 0.77 | **0.66** | 2.96 | 3.29 | 1.54 | 0.89 | 0.83 | 1.41 | 4.83 | 104.13 |
| | $W$+jets (CT14lo) | 90.42 | 1.08 | 1.60 | 4.56 | 4.64 | 3.38 | 1.60 | **0.56** | 3.05 | 23.67 | 94.95 |
| | $Z$+jets (CTEQ6L1) | 83.18 | **0.81** | 1.19 | 6.83 | 12.62 | 6.25 | 7.04 | 1.44 | 2.22 | 40.67 | 79.37 |
| $p_x$ | Leptoquark | 20.26 | **0.21** | 0.26 | 0.95 | 1.39 | 0.73 | 0.25 | 0.41 | 0.43 | 10.53 | 5.89 |
| | $t\bar{t}$ (CT14lo, Vincia) | 15.40 | 0.19 | **0.13** | 0.65 | 1.03 | 0.82 | 0.31 | 0.41 | 0.30 | 1.00 | 11.22 |
| | $W$+jets (CT14lo) | 11.84 | 0.26 | **0.21** | 1.07 | 0.90 | 0.75 | **0.21** | 0.24 | 0.22 | 8.75 | 11.04 |
| | $Z$+jets (CTEQ6L1) | 10.06 | 0.23 | **0.19** | 1.19 | 0.66 | 1.22 | 1.07 | 0.35 | 0.31 | 16.08 | 10.74 |

Table 2: Result of data recovery performances on 4 unseen physics distributions. We report the 1-D Wasserstein distance between the truth-level data and detector-level data / recovered data via various methods for $p_T, E$ and $p_x$ (complete results in Sec. B.4). The best results are noted in red.

### 3.4 IMAGE INVERSION OF MNIST DIGITS MIXTURE

In this section, we apply the proposed methods to a high-dimensional image EIP. The images of MNIST digit "9" continuously transform into MNIST digit "6" over time $t \in [0, 1]$. The images are all "9" at $t = 0$ and become "6" at $t = 1$. For $0 < t < 1$, the images are mixtures of the two digits,

resembling "6" more and "9" less as $t$ approaches 1. Details of the process of generating the digits are described in Sec. B.6. At the inference stage, given a set of blurred images, which come from a common prior at an unknown interpolation time $t$, the objective is to recover the corresponding clean images for each blurred image in the set.

Let $x_{a,b}$ denote the $a$-th rows' $b$-th pixel value in a MNIST image $x \in \mathbb{R}^{28 \times 28}$. The images are blurred in an element-wise way, and the forward process is given as,

$$y_{a,b} = x_{a,b} + n(x_{a,b}), \quad n(x_{a,b}) \sim \begin{cases} \delta(-x_{a,b}), & \text{with probability } 0.9; \\ \mathcal{N}(0,2), & \text{with probability } 0.1. \end{cases} \tag{5}$$

Setting $N = 128$ and $\phi_w : \mathbb{R}^{128 \times 28 \times 28} \to \mathbb{R}^{28 \times 28}$ for EI-DDPM and EI-FM, we compare our proposed methods with cFM, DDPM, and SBUnfold for the image inversion task. Each method is provided with pairs of clean images and blurred images resulting from priors with $t \in [0.1, 0.4] \cup [0.6, 0.9]$. At the inference time, each method aims to recover the original images from a set of images with the same but unknown $t$.

First, we visualize the recovery performance for $t = 0.5$ in Fig. 5. We can observe that EI-FM and EI-DDPM capture the structure of the truth more precisely. While other baselines' recoveries have visually greater differences with the truth's structures. Then we sweep $t \in [0, 1]$ with an interval 0.01 and evaluate the pixel-wise mean squared error (MSE) and structural similarity index measure (SSIM) between the recovered images and the truth for each method. Results in Fig. 6 shows EI-FM and EI-DDPM's superior performance in both MSE and SSIM, indicating that EI-FM and EI-DDPM can scale up to high-dimensional settings and effectively incorporate the ensemble information for posterior inferences and generalizations.

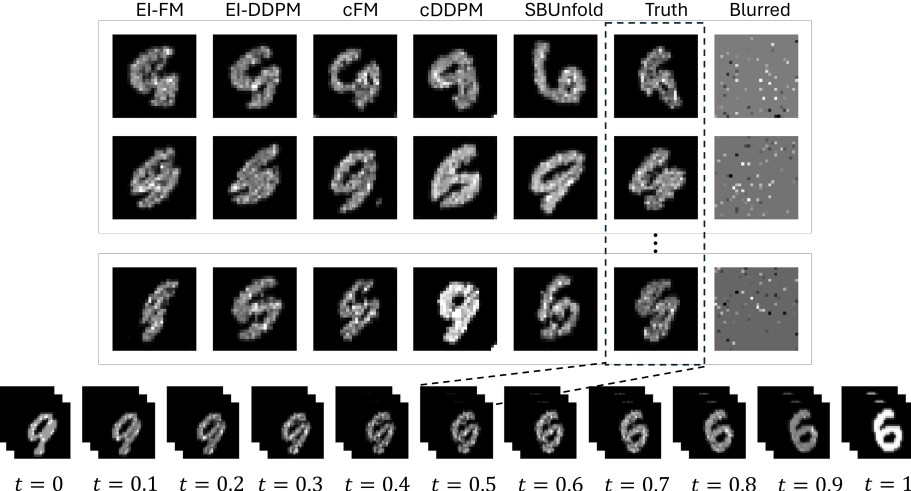

Figure 5: Upper: the recovered images via different methods, the truth ($t = 0.5$), and the blurred images. Lower: the transformation process from digit "9" to "6".

## 4 CONCLUSIONS AND FUTURE DIRECTIONS

We introduce EIP, in which one aims to invert for an ensemble that is distributed according to the pushforward of a prior under a forward process. To address this problem, we propose a posterior sampling framework, i.e., the ensemble inverse generative model, that is conditioned on both the measurements and the ensemble information extracted from an observation set via a permutation invariant NN. The proposed EI-DDPM and EI-FM demonstrate superior posterior inference and generalization abilities across several cases, including unfolding and inverse imaging. Future research directions include provable guarantees on the discrepancy between the recovered distributions and the prior, and optimal structures for ensemble information extraction.

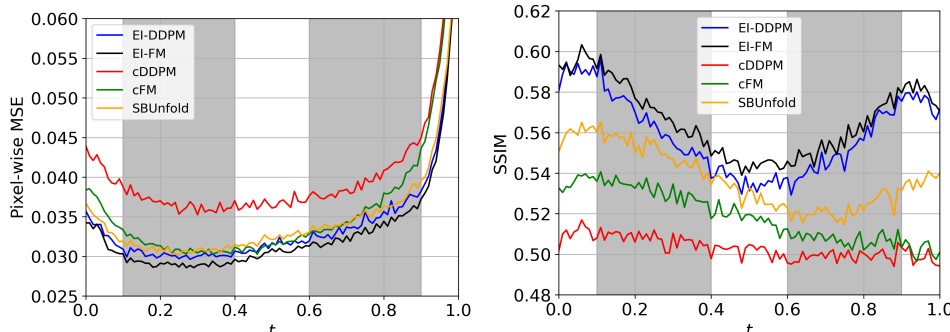

Figure 6: Pixel-wise MSE(↓) and SSIM(↑) in latent space vs. $t$. Grey areas denote the priors that are included in the training data.

## 5 ETHICS STATEMENT

This work does not raise any specific ethical concerns.

## 6 REPRODUCIBILITY STATEMENT

We provide a detailed description of the experiment implementation in Sec. B in the appendix. We also provide the code in the supplementary materials. We will publish the code on GitHub if this paper is accepted.

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

## A AN INTRODUCTION TO DDPM AND FM

Here we provide a brief introduction for the conditional version of DDPM and FM.

**DDPM:** DDPM learns to reverse a forward noising process and generate data by applying the learned reverse process to map samples from a Gaussian distribution $q_0 = \mathcal{N}(\mathbf{0}, \mathbf{I})$ to the target distribution $q_1$. In the forward process, a sample starting from $x_0 \sim q_1$ is gradually corrupted:

$$q(x_t|x_{t-1}) = \mathcal{N}(x_t, \sqrt{1-\beta_t}x_t, \beta_t\mathbf{I}), \quad t = 1, \cdots, T, \tag{6}$$

in which $T$ is the number of total steps and $\beta_1, \cdots, \beta_T$ are pre-defined schedules. $x_T \sim \mathcal{N}(\mathbf{0}, \mathbf{I})$ when $T$ is sufficiently large. Next, with $\alpha_t := 1 - \beta_t$ and $\bar{\alpha}_t := \prod_{s=1}^t \alpha_s$, DDPM models the reverse process as

$$p_\theta(x_{t-1}|x_t, z) = \mathcal{N}(x_{t-1}; \mu_\theta(x_t, t, z), \sigma_t^2\mathbf{I}), \quad \mu_\theta(x_t, t, z) = \frac{1}{\sqrt{\alpha_t}}\left(x_t - \frac{\beta_t}{\sqrt{1-\bar{\alpha}_t}}\varepsilon_\theta(x_t, t, z)\right) \tag{7}$$

in which $z$ is the conditional information, which is a function of $y, \mathcal{Y}$ in EIP-II. $\varepsilon_\theta$ is a neural network(NN) parameterized with $\theta$, and $\sigma_t^2$ is the variance in the reverse process derived from the forward process. With the objective of minimizing the expected MSE between a noise $\varepsilon \sim \mathcal{N}(\mathbf{0}, \mathbf{I})$ and the model's prediction, i.e.,

$$\arg\min_\theta \mathbb{E}_{x_0, z, t, \varepsilon}[\|\varepsilon_\theta(x_t, t, z) - \varepsilon\|^2], \quad x_t = \sqrt{\bar{\alpha}_t}x_0 + \sqrt{1-\bar{\alpha}_t}\varepsilon, \tag{8}$$

DDPM model learns the reverse process in equation 7.

**FM:** FM aims to learn continuous flows between an initial distribution $q_0$ and the target distribution $q_1$ by learning the velocity fields across time. Consider $d$-dimensional data, define a stochastic process $x_t = \Psi_t(x_0, x_1) : [0, 1] \times \mathbb{R}^d \times \mathbb{R}^d \to \mathbb{R}^d$ with $x_0 \sim q_0$ and $x_1 \sim q_1$ that are twice differentiable in space and time and uniformly Lipschitz in time satisfying $\Psi_0(x_0, x_1) = x_0$, $\Psi_1(x_0, x_1) = x_1$. The velocity field is defined via $v^\Psi(x, t) = \mathbb{E}[\frac{d}{dt}\Psi_t|X_t = x]$. FM aims to learn the velocity field with an NN $\varepsilon_\theta(x_t, t, z)$ parameterized by $\theta$. Similarly, $z$ is the conditional information, which stands for a function of $y, \mathcal{Y}$ in EIP-II. FM's objective minimize the MSE between the $v^\Psi(x, t)$ and $\varepsilon_\theta(x_t, t, z)$. Although $v^\Psi(x, t)$ is intractable since it is an average over all possible trajectories crossing $x$, one can optimize the objective via the following equivalence Lipman et al. (2023),

$$\arg\min_\theta \int_0^1 \mathbb{E}[\|\varepsilon_\theta(x_t, t, z) - v^\Psi(x_t, t)\|^2]dt = \arg\min_\theta \int_0^1 \mathbb{E}[\|\varepsilon_\theta(x_t, t, z) - \frac{d}{dt}\Psi_t(x_0, x_1)\|^2]dt, \tag{9}$$

where we recall $x_t = \Psi_t(x_0, x_1)$. Note that $\Psi_t(x_0, x_1)$ can be picked by the user. One common and simple choice is the linear interpolants $\Psi_t(x_0, x_1) = tx_1 + (1-t)x_0$, with $\frac{d}{dt}\Psi_t(x_0, x_1) = x_1 - x_0$, leading to a concrete objective in equation 9 that can be efficiently estimated via Monte-Carlo.

## B EXPERIMENT DETAILS

### B.1 MODEL CONFIGURATION

All experiments are run on an NVIDIA L40 GPU with 46 GB memory. The configuration for each experiment is described as follows.

**2-D Gaussian EIP:** $\phi_w : \mathbb{R}^{4000 \times 2} \to \mathbb{R}^3$ is implemented according to Lee et al. (2019) and consists of an encoder using a single Induced Set Attention Block (ISAB) encoder to capture set-element interactions with linear-time attention via trainable inducing points, and a decoder that performs Pooling by Multihead Attention (PMA), followed by a Set Attention Block (SAB) to model correlations among the pooled outputs, and a final linear projection to the 3-D ensemble information. Specifically, ISAB, which uses multihead attention with 4 heads, takes an unordered set $\mathcal{Y} \in \mathbb{R}^{4000 \times 2}$ as the input and maps the input to 128-D embeddings. PMA and SAB both apply multihead attention with 4 heads and have embedding sizes of 128. The final linear projection is a linear layer mapping from 128-D embeddings to 3-D ensemble information.

$\varepsilon_\theta$ for EI-DDPM and EI-FM consists of Multi-Layer Perceptrons (MLPs), incorporating a time embedding. The network first takes the concatenation of intermediate data $x_t \in \mathbb{R}^2$, the single measurement $y \in \mathbb{R}^2$, and the ensemble information $\phi_w(\mathcal{Y}) \in \mathbb{R}^3$ as the input and processes it through a 64-unit hidden layer. The outputs are added with a learned time embedding with time $t$ as an input, and then processed through 64-unit hidden layers. Skip connections are employed between the input and output of the main block. The final outputs are 2-D variables representing the predicted noise / velocity field at time $t$ for EI-DDPM / EI-FM. EI-DDPM has a total number of steps $T = 100$. The noise schedule is defined linearly from an initial noise level of $\beta_1 = 1 \times 10^{-4}$ to a final noise level of $\beta_T = 0.02$ across timesteps $t = 1, \dots, T$. The discretization interval for EI-FM during inference time is set as $\Delta t = 0.01$.

**Particle Physics Data Unfolding:** $\phi_w : \mathbb{R}^{2000 \times 7} \to \mathbb{R}^6$ shares the same structure as in 2-D Gaussian EIP, with input and output dimension adapted. $\varepsilon_\theta$ also share similar structures as in 2-D Gaussian EIP, with the number of units in hidden layers changed. The input of the concatenation of intermediate data $x_t \in \mathbb{R}^7$, the single measurement $y \in \mathbb{R}^7$, and the ensemble information $\phi_w(\mathcal{Y}) \in \mathbb{R}^6$ first goes through a 256-unit hidden layer and the added with a learned time embedding. The subsequent layers for mapping into 7-D noise / velocity field consist of 256-unit and 512-unit linear layers. The total time steps for EI-DDPM is set as $T = 500$, and noise schedule for EI-DDPM remains the same as in 2-D Gaussian EIP. The discretization interval for EI-FM during inference time is set as $\Delta t = 0.002$.

**Image inversion of MNIST Digits Mixture:** The structures of $\varepsilon_\theta$ and $\phi_w$ are modified to facilitate processing images in this case. For a set of images, $\phi_w : \mathbb{R}^{128 \times 28 \times 28} \to \mathbb{R}^{28 \times 28}$ first process each image in $\mathcal{Y}$ with a four-stage convolutional encoder with $3 \times 3$ convolution kernels for image feature representations. The representation for each image is flattened into 128-dim variables. Then the representation set is mapped into the ensemble information $\phi_w(\mathcal{Y}) \in \mathbb{R}^{28 \times 28}$ via a set transformer with the same structure as in 2-D Gaussian EIP (input and output dimension adapted).

$\varepsilon_\theta$ employs an U-net structure Ronneberger et al. (2015), which accepts a matrix of 3 channels and time $t$ as inputs. The 3 channels in the matrix are $x_t \in \mathbb{R}^{28 \times 28}$, $y \in \mathbb{R}^{28 \times 28}$ and $\phi_w(\mathcal{Y}) \in \mathbb{R}^{28 \times 28}$. The final outputs are $\mathbb{R}^{28 \times 28}$ variables representing the predicted noise / velocity field at time $t$ for EI-DDPM / EI-FM. The total time steps for EI-DDPM is set as $T = 500$, and the noise schedule for EI-DDPM remains the same as in 2-D Gaussian EIP. The discretization interval for EI-FM during inference time is set as $\Delta t = 0.002$.

## B.2 EFFECT OF $N, N'$ IN 2-D GAUSSIAN EIP

Note that we assume $N' \gg N$ in most cases, i.e., the available sample number is sufficient to form observation sets that can contain the ensemble information. Fixing the observation set size $N$ for training can contribute to a simpler training pipeline and a more stable optimization process. And this does not impact the inference since size $N$ observation sets are available. However, a fixed $N$ for training is not strictly required. If the number of available observations for inference stays close to $N$, and yet is not fixed, we recommend that users employ random $N$s within a range aligning with the inference requirements during training. In this way, the inference algorithm can automatically work for changeable set sizes, as in the training range.

Next we numerically investigate the effect of $N$ and $N'$ under the setting of 2-D Gaussian inverse problem in Sec. 3.2. First for the effect of $N$, we train EI-FM with observation set size $N$ from 5 to 32000. The results in Fig. 7a show that for small $N \le 10$, the recovery performance is even worse than the baseline cFM without any group information. $\mathcal{Y}$ with too small set sizes cannot represent the ensemble information and even mislead the model in both training and inference. As $N$ grows larger, EI-FM displays its advantage over cFM by leveraging valid ensemble information from $\mathcal{Y}$. The recovery performance evaluated by SWD increases with the growth of $N$ and stabilizes when $N$ reaches a sufficiently large value.

Next we consider the cases such that the number of samples to recover $N'$ is smaller than $N$. Take an EI-FM model trained with $N = 4000$. Assuming that only $N'$ samples are available during the inference time, these $N'$ samples are duplicated until the set has $N$ samples to perform Alg. 2. To evaluate the SWD metric, this process is repeated several times until 40000 samples are recovered. The results shown in Fig. 7b indicate that SWD decreases as $N'$ grows up to 4000. For $N'$ that

are not significantly less than $N$, such as $N' = 1000$, the duplication strategy can still yield an SWD close to the $N' = 4000$ case, since the sets after duplication can still effectively represent the ensemble information. Notably, even with $N'$ as small as 10, EI-FM slightly outperforms cFM, which performs a sample-wise recovery. This highlights the effect of ensemble information in EIP-II.

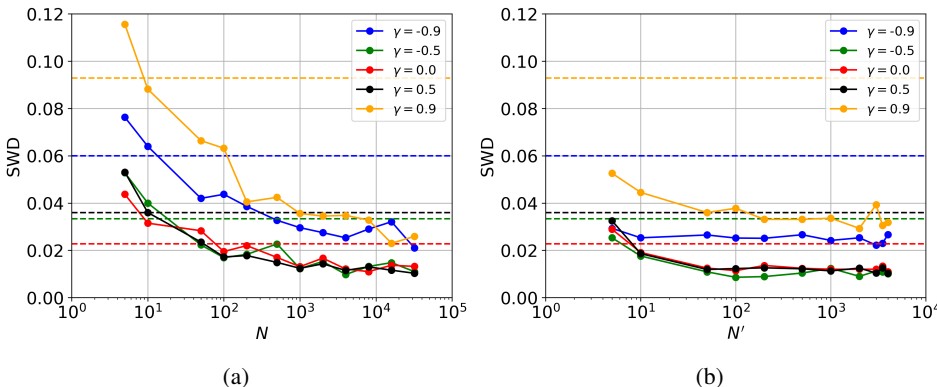

(a)                                                 (b)

Figure 7: Average SWD between the truth and the recovery vs. $\gamma$. The horizontal dashed lines represent the performance of cFM baselines. (a) is for EI-FM trained with different $N$, evaluated over $40000$ samples. We also provide the stats of cFM as a baseline. (b) is for EI-FM trained with $N = 4000$, evaluated over $40000$ samples. It is assumed that only $N'$ samples are available during the inference time and Alg. 2 is implemented via the duplication strategy.

### B.3  EXTENSION OF 2-D GAUSSIAN EIP

Here we present an extension of the 2-D Gaussian EIP, in which the number of parameters determining the prior increases from 1 to 3. Consider the prior

$$x|\gamma \sim \mathcal{N}\left(\begin{bmatrix} \mu_1 \\ \mu_2 \end{bmatrix}, \begin{bmatrix} 1 & \gamma_1 \\ \gamma_1 & 1 \end{bmatrix}\right), \quad \gamma = (\mu_1, \mu_2, \gamma_1), \tag{10}$$

in which $\mu_1, \mu_2, \gamma_1$ are 3 independent parameters. Samples from this prior undergo the same forward process as equation 4. One still aims to recover the prior given its observation set $\mathcal{Y}$ corresponding to an unknown $\gamma$.

During the training stage, truth-observation pairs resulting from priors with $\gamma_1 \in [-0.75, -0.25] \cup [0.25, 0.75]$ and $\mu_1, \mu_2 \in [-1.5, -0.5] \cup [0.5, 1.5]$ are provided. In the inference time, we evaluate the recovery performance for priors with $\gamma_1 \in [-1, 1]$ and $\mu_1, \mu_2 \in [-2, 2]$ perturbed by equation 4. We compare EI-FM with $\phi_w : \mathbb{R}^{4000 \times 2} \to \mathbb{R}^5$, cFM without any ensemble information and cFM-$\gamma$, which is directly provided with $\gamma = (\mu_1, \mu_2, \gamma_1)$. To illustrate the recovery performance vs. 3 parameters, we make 3-D figures, in which x,y axes stand for $\mu_1, \mu_2$ respectively, and each figure corresponds to a specific $\gamma_1$. The $z$ axis stands for the metric of measuring the distribution similarity, i.e., SWD. The results in Fig. 8 show that EI-FM can achieve comparable performances to cFM-$\gamma$ across all ranges of $\gamma = (\mu_1, \mu_2, \gamma_1)$ and achieves much better performances than cFM. EI-FM's close performance to cFM-$\gamma$ (with direct knowledge of the prior) further illustrates that EI-FM can still extract valid ensemble information for posterior inference and generalization as the number of parameters determining the prior increases.

### B.4  COMPLETE RESULTS OF PARTICLE PHYSICS DATA UNFOLDING

In this section, we present the complete result of the 1-D Wasserstein distance between the truth-level data and detector-level data / recovered data via various methods for all 7 components in the physics process in Table 3. The detector-level distortion for $\eta$ and $\phi$ is small, and their detector-level distributions have already come close to the true prior. Therefore, some best performances for $\eta, \phi$ appear in the detector-level data, i.e., before unfolding.

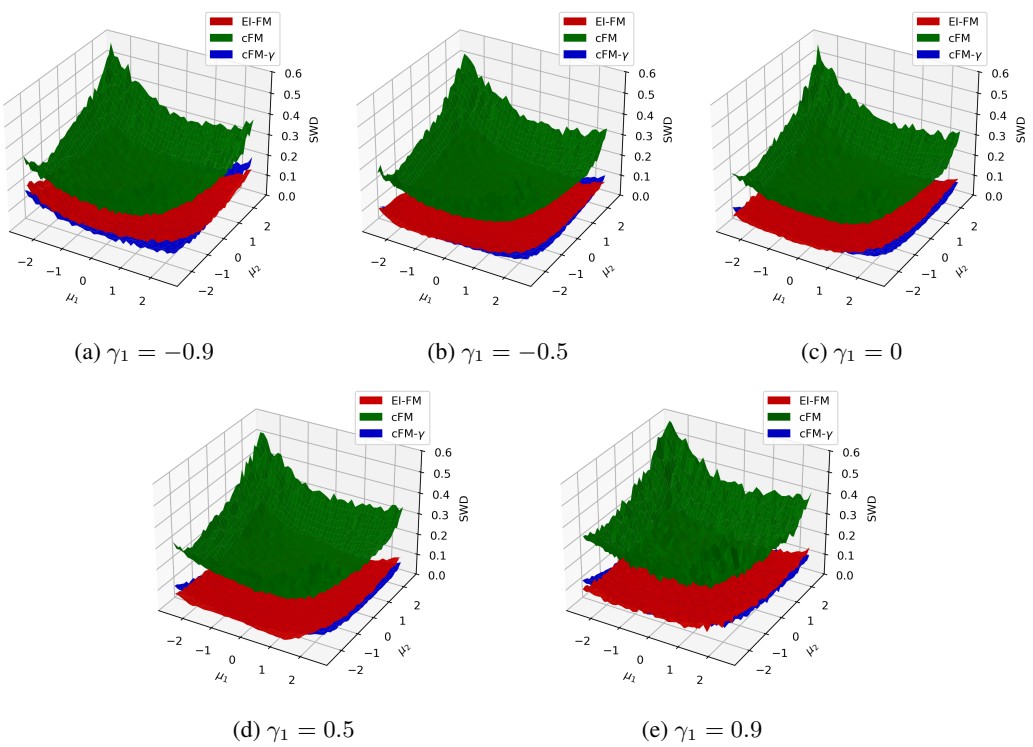

Figure 8: Average sample-wise SWD($\downarrow$) between the truth and the recovery vs. $(\mu_1, \mu_2)$ for $\gamma_1 = \{-0.9, -0.5, 0, 0.5, 0.9\}$, evaluated over $40000$ samples.

| WD ($\downarrow$) | Name | Detector | EI-DDPM | EI-FM | cDDPM | cFM | GDDPM-v | GDDPM | Omnifold-best | Omnifold-combine | SBUnfold | Sourcerer |
|---|---|---|---|---|---|---|---|---|---|---|---|---|
| $p_T$ | Leptoquark | 31.85 | 0.44 | 0.44 | 1.08 | 2.65 | 0.73 | 0.44 | **0.19** | 0.82 | 18.10 | 7.96 |
| | $t\bar{t}$ (CT14lo, Vincia) | 24.18 | 0.44 | **0.23** | 1.01 | 3.36 | 1.36 | 0.55 | 0.60 | 0.52 | 1.88 | 15.14 |
| | $W$+jets (CT14lo) | 18.60 | 0.60 | 0.44 | 2.41 | 4.14 | 0.53 | 0.48 | **0.11** | 0.37 | 21.07 | 25.96 |
| | $Z$+jets (CTEQ6L1) | 15.81 | 0.51 | **0.45** | 2.55 | 5.55 | 2.25 | 1.98 | 0.48 | 0.64 | 25.18 | 19.59 |
| $\eta$ | Leptoquark | 0.00074 | 0.00079 | 0.00096 | 0.00182 | 0.00272 | 0.00255 | **0.00056** | 0.01936 | 0.00758 | 0.04350 | 0.02079 |
| | $t\bar{t}$ (CT14lo, Vincia) | 0.00080 | 0.00075 | 0.00095 | 0.00128 | 0.00298 | 0.00363 | **0.00071** | 0.00689 | 0.00979 | 0.03795 | 0.01997 |
| | $W$+jets (CT14lo) | **0.00060** | 0.00096 | 0.00109 | 0.00186 | 0.00406 | 0.00375 | 0.00080 | 0.01945 | 0.01596 | 0.06352 | 0.00728 |
| | $Z$+jets (CTEQ6L1) | **0.00065** | 0.00093 | 0.00111 | 0.00202 | 0.00466 | 0.00298 | 0.00072 | 0.00934 | 0.02069 | 0.07880 | 0.03140 |
| $\phi$ | Leptoquark | 0.00140 | 0.00091 | **0.00069** | 0.00662 | 0.00379 | 0.00342 | 0.00142 | 0.01492 | 0.00534 | 0.01452 | 0.01891 |
| | $t\bar{t}$ (CT14lo, Vincia) | 0.00144 | 0.00096 | **0.00078** | 0.00718 | 0.00381 | 0.00383 | 0.00158 | 0.00609 | 0.00397 | 0.01493 | 0.01137 |
| | $W$+jets (CT14lo) | 0.00153 | 0.00092 | **0.00074** | 0.00803 | 0.00401 | 0.00373 | 0.00159 | 0.00689 | 0.00625 | 0.01426 | 0.01482 |
| | $Z$+jets (CTEQ6L1) | 0.00153 | 0.00107 | **0.00071** | 0.00836 | 0.00427 | 0.00396 | 0.00177 | 0.03053 | 0.00388 | 0.01552 | 0.03828 |
| $E$ | Leptoquark | 83.87 | **0.46** | 0.76 | 4.70 | 2.66 | 1.47 | 0.63 | 1.08 | 0.47 | 13.99 | 57.42 |
| | $t\bar{t}$ (CT14lo, Vincia) | 69.70 | 0.77 | **0.66** | 2.96 | 3.29 | 1.54 | 0.89 | 0.83 | 1.41 | 4.83 | 104.13 |
| | $W$+jets (CT14lo) | 90.42 | 1.08 | 1.60 | 4.56 | 4.64 | 3.38 | 1.60 | **0.56** | 3.05 | 23.67 | 94.95 |
| | $Z$+jets (CTEQ6L1) | 83.18 | **0.81** | 1.19 | 6.83 | 12.62 | 6.25 | 7.04 | 1.44 | 2.22 | 40.67 | 79.37 |
| $p_x$ | Leptoquark | 20.26 | **0.21** | 0.26 | 0.95 | 1.39 | 0.73 | 0.25 | 0.41 | 0.43 | 10.53 | 5.89 |
| | $t\bar{t}$ (CT14lo, Vincia) | 15.40 | 0.19 | **0.13** | 0.65 | 1.03 | 0.82 | 0.31 | 0.41 | 0.30 | 1.00 | 11.22 |
| | $W$+jets (CT14lo) | 11.84 | 0.26 | **0.21** | 1.07 | 0.90 | 0.75 | **0.21** | 0.24 | 0.22 | 8.75 | 11.04 |
| | $Z$+jets (CTEQ6L1) | 10.06 | 0.23 | **0.19** | 1.19 | 0.66 | 1.22 | 1.07 | 0.35 | 0.31 | 16.08 | 10.74 |
| $p_y$ | Leptoquark | 20.29 | **0.25** | **0.25** | 0.95 | 1.63 | 0.36 | **0.25** | 0.53 | 0.56 | 10.81 | 7.41 |
| | $t\bar{t}$ (CT14lo, Vincia) | 15.39 | 0.23 | **0.13** | 0.90 | 1.47 | 0.65 | 0.31 | 0.45 | 0.27 | 0.89 | 10.80 |
| | $W$+jets (CT14lo) | 11.84 | 0.28 | **0.18** | 1.51 | 1.54 | 0.28 | 0.19 | 0.22 | 0.26 | 8.86 | 14.32 |
| | $Z$+jets (CTEQ6L1) | 10.06 | 0.25 | **0.19** | 1.87 | 1.44 | 1.28 | 1.09 | 0.52 | 0.27 | 15.72 | 11.77 |
| $p_z$ | Leptoquark | 70.72 | 0.67 | **0.56** | 6.78 | 5.04 | 0.99 | 0.86 | 3.15 | 1.00 | 17.94 | 45.64 |
| | $t\bar{t}$ (CT14lo, Vincia) | 60.38 | 0.86 | **0.52** | 6.22 | 4.21 | 0.87 | 1.06 | 2.09 | 2.28 | 11.53 | 74.01 |
| | $W$+jets (CT14lo) | 84.96 | **1.18** | 1.41 | 7.48 | 5.82 | 3.83 | 1.57 | 4.25 | 3.52 | 12.38 | 70.12 |
| | $Z$+jets (CTEQ6L1) | 78.70 | **1.06** | 1.15 | 6.50 | 7.09 | 5.89 | 6.89 | 2.90 | 2.96 | 33.05 | 62.07 |

Table 3: Result of data recovery performances on 4 unseen physics distributions. We report the 1-D Wasserstein distance between the truth-level data and detector-level data / recovered data via various methods for $p_T, \eta, \phi, E, p_x, p_y, p_z$. The best results are noted in red.

## B.5   TARP COVERAGE AS AN ADDITIONAL METRIC FOR POSTERIOR ACCURACY

To further assess the accuracy of posterior samplers, we employ the Test of Accuracy with Random Points (TARP) expected coverage metric that is introduced in Lemos et al. (2023). Let $(x^*, y)$ denote the truth-observation pairs, TARP assesses whether an estimated posterior $\hat{p}(y|x)$ correctly

approximates the true posterior $p(y|x)$ by examining how often credible regions constructed from $\hat{p}$ contain the truth $x^*$. For each truth-observation pair $(x^*, y)$, a reference point $x_r$ is randomly drawn, and the TARP region is defined as

$$D_{x_r}(\hat{p}, \alpha, y, d) = \{x : d(x, x_r) \leq d(x^*, x_r)\}, \quad (11)$$

in which $d(\cdot, \cdot)$ is a distance function and $1 - \alpha$ is the credibility level. The expected coverage for credibility level $1 - \alpha$ is

$$\text{ECP}(\hat{p}, \alpha, D_{x_r}) = \mathbb{E}_{(x^*, y)}[\mathbf{1}\{x^* \in D_{x_r}(\hat{p}, \alpha, y)\}]. \quad (12)$$

It is proven in Lemos et al. (2023) that matching the identity $\forall \alpha \in (0, 1), \text{ECP}(\hat{p}, \alpha, D_{x_r}) = 1 - \alpha$ indicates that $\hat{p}$ is the true posterior $p$. In practice, one evaluates this metric by scanning over credibility levels $1 - \alpha$ and plotting the ECP vs. credibility level curve for examining the posterior's correctness. For each curve, we also let $\alpha$ follow a uniform distribution in $(0, 1)$ and report $e = \mathbb{E}_\alpha[|\text{ECP}(\hat{p}, \alpha, D_{x_r}) - (1 - \alpha)|]$ for each estimated posterior $\hat{p}$ to measure its deviation from the ideal case.

The ECP vs. credibility level results for various posterior samplers in the 2-D Gaussian EIP at Sec. 3.2 and in the HEP unfolding at Sec. 3.3 are shown in Fig. 9 and Fig. 10 respectively. While all posterior samples' ECP vs. credibility level curves are close to the ideal case, those with ensemble information (e.g., EI-FM, EI-DDPM, cFM-$\gamma$, GDDPMs) display slightly smaller deviations $e$ from the ideal case, illustrating the effectiveness of ensemble information in modeling posteriors.

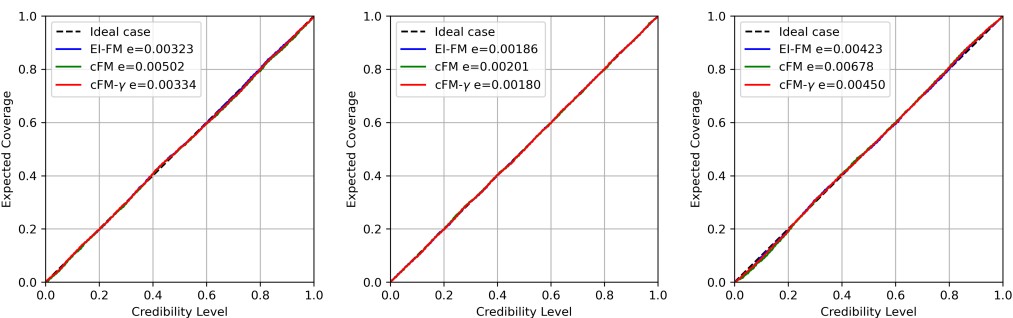

Figure 9: ECP vs. credibility level curves for different models in the 2-D Gaussian EIP at Sec. 3.2. Left: $\rho = -0.9$. Middle: $\rho = 0$. Right: $\rho = 0.9$.

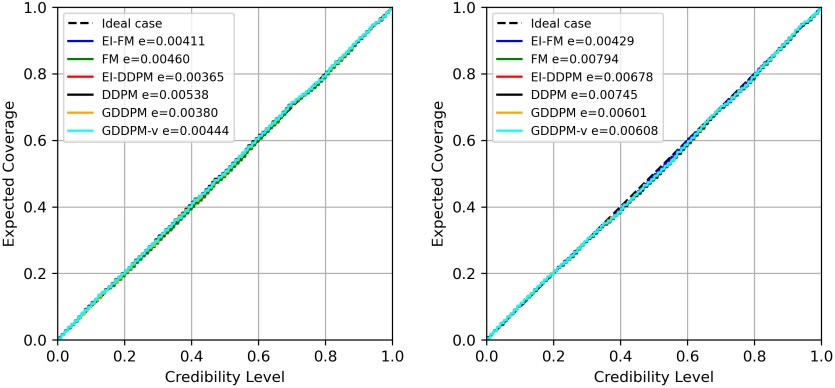

Figure 10: ECP vs. credibility level curves for different models in the particle physics unfolding task at Sec. 3.3. Left: Leptoquark process. Right: $t\bar{t}$ (CT14lo, Vincia) process.

### B.6    PROCESS OF GENERATING MNIST DIGITS MIXTURE

The process of creating the mixture of two MNIST digits following Haviv et al. (2025) is described as follows. First, the MNIST digit images are converted to point clouds. Then an entropically regularized optimal transport (OT) plan between two weighted point clouds is computed using OTT's

Sinkhorn solver, producing a soft matching matrix. Based on the matrix, greedy "rounded match-ing" is applied by repeatedly selecting the maximum probability entry in the matrix, assigning that source to the corresponding target, and zeroing out the associated row and column to prevent reuse. This process iterates until all points are matched, leading to a permutation-like hard assignment that approximates the true optimal permutation matrix implied by the OT solution. The resulting hard assignment defines a transport path parameterized by time $t$, where $t = 0$ corresponds to the initial point clouds and $t = 1$ corresponds to the target point clouds. The intermediate $t$ interpolates each point along its assigned displacement toward its target. Finally, the point clouds are converted back to images.

