# OpenReview forum: "The Ensemble Inverse Problem: Applications and Methods"
_ICLR.cc/2026/Conference — Submitted to ICLR 2026_

### Official Review · Reviewer_2G23 · 2025-10-31

**Soundness:** 2
**Presentation:** 2
**Contribution:** 1
**Rating:** 2
**Confidence:** 2

**Summary:**

The paper introduces the Ensemble Inverse Problem (EIP) framework, in which the goal is to infer an unknown variable from a measurement without explicit access to the forward model or prior distribution. The authors describe two settings, EIP-I (recover prior) and EIP-II (recover posterior). The paper focuses on EIP-II and proposes ensemble inverse generative models (EI-DDPM and EI-FM) that condition posterior sampling on both a single observation and an observation ensemble. To extract information from the observation set, the method uses a permutation-invariant set encoder, such as Deep Sets or the Set Transformer. Experiments include 2-D Gaussian toy distributions, high-energy physics unfolding, and an MNIST image inversion benchmark. The authors claim improved generalization to unseen priors and avoidance of iterative inference.

**Strengths:**

- The paper introduces a clear formalization of EIP and distinguishes EIP-I and EIP-II, clarifying the problem space and highlighting a setting where the forward operator is unknown.
- The method uses established conditional generative models (DDPM and flow matching) and extends them to incorporate set-based ensemble conditioning.
- A permutation-invariant representation is included to encode the ensemble of observations, aligning with prior set-modeling work.
- Experiments across multiple synthetic and real-world tasks demonstrate feasibility (2-D Gaussian, HEP unfolding, MNIST inversion).

**Weaknesses:**

- The method is largely a conditional generative model with an added set encoder. The paper frames this as a new problem, but the algorithmic contribution is incremental, in my opinion.
- The practical relevance of EIP is not strongly justified. Real-world cases where forward operators are truly unavailable yet training observations exist remain unclear.
- Toy Gaussian and MNIST blur/noise settings do not convincingly demonstrate scalability or relevance for complex inversion tasks.
- No comparison to stronger modern inference baselines (e.g., existing diffusion-based inverse samplers outside EIP literature).
- Duplicating samples when $N’<N$ is a rather ad-hoc approach.
- Claims of generalization to unseen priors are qualitative and lack theoretical justification.
- No ablations isolating the benefit of set conditioning versus simply conditioning on summary statistics (moments).

**Questions:**

- Can you provide some concrete, real-world settings where the forward operator is unknown, but large paired datasets exist for training?
- How does performance scale with dimensionality beyond MNIST? Are there plans to test the proposed algorithms on natural images or scientific imaging?
- What specific features of the set encoder contribute to generalization to unseen priors beyond simple moment-based approaches?

---

> ### Author Response · Authors · 2025-11-18
>
> 1. The main novelty of the proposed methods lies in how to use the information contained in the observation set to help construct the posterior and the extension from sample-wise posterior sampler into distributional-dependent posterior sampler. As far as we know, this is the first work focusing on incorporating the  ensemble information  extracted by neural networks into the modeling of the posterior.
>
> 2. The high-energy physics (HEP) unfolding task is a major real-world application and motivation behind this work.  It is a widely studied practical problem in HEP, with significant attention in both methodology and applications, as shown in 1911.09107, 2406.01507 and 2308.1235.
>
> 3. To further display the effectiveness in real world settings, we plan to add a real-world audio inverse problem and update the results by Dec 03.
>
> 4. We have considered a comprehensive range of baselines in the unfolding task, which is a well-studied problem in HEP.
>
>     The EIP setting is less studied in the image domain, and as mentioned in the first question, this is the first work focusing on incorporating the  ensemble information  extracted by neural networks into modeling of the posterior. Therefore, we compare with vanilla conditional diffusion models and an existing work SBUnfold.
>
>     We kindly request the reviewer to please share any specific works they consider more appropriate for baselines for EIP tasks.
>
> 5. In common cases $N’\geq N$,  duplicating samples is only a remedy at inference time when provided with insufficient samples. Alg. 2 requires at least $N$ samples, and the duplication is an effective and simple way to create a set of $N$ observation samples if the provided samples are insufficient. As shown in Fig. 7(b), even with $N’=100$ and $N=4000$, the duplication strategy performs better than the baseline without the ensemble information.
>
> 6. In traditional regression, a model generalizes because it learns a functional relationship that holds for all samples drawn from the same underlying distribution, rather than memorizing specific inputs. This work extends the same principle to the **distributional level**: instead of mapping individual samples to predictions, EI models map an observation distribution to its corresponding **posterior distribution** $p(x∣y,\mathcal{Y})$. Since the model learns a stable distribution-to-distribution relationship—analogous to learning a sample-to-sample function in regression—it naturally generalizes to unseen observation sets drawn from the forward operator and a prior family. We will add this intuition into the paper.
>
> 	Rigorous theoretical proofs are indeed useful to develop bounds on the performance of the proposed methods and are promising future directions. Nevertheless, at present, this development is beyond the scope of this work.
>
>
> 7. We explicitly compare conditional on moments (GDDPM, GDDPM-v) with our methods in the 2nd and 3rd paragraph of Sec. 3.3 (line 375-391), which we summarize as follows.
>
>     “GDDPM implicitly assumes that $p_T$ contains the complete distributional information of the $7$-component vector. While GDDPM-v does not have this assumption by incorporating the moments of all components. EI-DDPM/FM also do not have this assumption since the input is the observation set instead of data of a single component. The performance order is: EI-DDPM/FM>GDDPM>GDDPM-v, reflecting that EI-DDPM/FM is the most adaptive and effective approach.”
>
> Questions:
>
> 1. The HEP unfolding task shown in the paper is a major real-world task fitting in this setting. We plan to additionally add a real-world audio inverse problem and update the results by Dec 03.
>
> 2. We agree that MNIST’s complexity is below the real-world images. We will further investigate the scalability for more high-dimensional EIP settings.
>
> 3. The ensemble information (features) produced by the permutation invariant structure  cannot be directly interpreted as statistical values such as moments. It is the information that the model learned to help build the posterior. For example in the toy Gaussian example, the ensemble information is a 3-dim vector and we provided it for $\gamma = \{-0.9,-0.5,0,0.5,0.9\}$:
>
>     $\gamma = -0.9$: [-1.2975,  0.2756,  0.5768]
>
>     $\gamma = -0.5$: [-1.1671,  0.6163,  0.2411]
>
>     $\gamma = 0$: [-1.0793,  0.6567, -0.0512]
>
>     $\gamma = 0.5$: [-0.9931,  0.6833, -0.3324]
>
>     $\gamma = 0.9$: [-0.8159,  0.0700, -1.2243]
>
>     We observe that when $\gamma$ changes from $-0.9$ to $0.9$, each component of the 3-dim ensemble information also changes in a descent or ascent order. This shows that the vectors successfully encode the distributional information to reconstruct the posterior.

---

> > ### Comment · Reviewer_2G23 · 2025-11-21
> >
> > I thank the authors for their detailed response.
> >
> > "We explicitly compare conditional on moments (GDDPM, GDDPM-v) with our methods...". I think I should have been more precise in my comments, all of which pertain to the imaging experiments.
> >
> > "We kindly request the reviewer to please share any specific works they consider more appropriate for baselines for EIP tasks...". The following two papers by Chung et al. are important baselines for imaging inverse problems.
> > 1. Diffusion posterior sampling (https://arxiv.org/abs/2209.14687)
> > 2. Improving Diffusion Models for Inverse Problems using Manifold Constraints (https://arxiv.org/abs/2206.00941)
> > Another important and related baseline is "Palette: Image-to-Image Diffusion Models" by Saharia et al. (https://arxiv.org/abs/2111.05826).
> >
> > While it might be impractical to ask for a comprehensive comparison with all these methods, at least a small discussion is warranted to put the proposed method in perspective.
> >
> > I would like to retain my original score, primarily because I am concerned about the novelty (and it is entirely possible that I have missed something). In my opinion, the paper is essentially a new way of conditional sampling with a set encoding (which is interesting but not strong enough to merit publication in a top-tier venue like ICLR).
> >
> > P.S.: I look forward to seeing the new numerical results that you wrote about in your response.

---

> > > ### Author Response · Authors · 2025-11-21
> > >
> > > 1. We clarify that GDDPM and GDDPM-v come from 2406.01507, which focuses on only the unfolding problem in HEP. Moments is an important statistics in jet kinematic information, therefore, moments are selected as the additional conditional information. It is not clear what the corresponding statistic is in the image EIP. Hence GDDPM cannot be applied to image EIP and this baseline is not included. While EI-DDPM/FM can automatically extract the ensemble information and serve as a general solution to EIP in different domains.
> > >
> > > 2. We thank the reviewer for the recommended references.
> > >
> > >     (1) 2209.14687: 2209.14687 explicitly requires an available forward operator, which slightly deviates from our setting which assumes that the forward operator is unknown. Nevertheless, as reviewer Ay8h points out, a simulator of the forward operator can be obtained via truth-observation pairs and approaches requiring available forward operators can be applied afterwards. Therefore, we plan to add a baseline following this procedure, i.e., first, train a simulator of the forward operator, second, couple with the approach in 2209.14687/2402.07808 for inversion.
> > >
> > >
> > >     (2) 2111.05826: 2111.05826 proposes an image inversion diffusion model that can solve a wide range of inverse problems—including colorization, inpainting, uncropping, and JPEG artifact restoration—within a unified framework. The wide range of inverse problems correspond to a wide range of forward operators, instead to a wide range of priors setting in this work. Moreover, this work focuses on tasks, in which single observations cannot carry the complete task information, e.g., from a single jet kinematic vector, one is unable to determine which physics process it comes from. However, from a single observation as considered in 2209.14687, the model can tell which task it is dealing with, making the ensemble information unnecessary. Therefore 2209.14687 is only weakly related to this work and may not serve as a baseline.
> > >
> > > 3. Regarding the novelty, we would like to emphasize that this work’s contribution lies in the formalization of EIP—an existing but previously undefined and insufficiently explored problem. Building on this formalization, we further provide general solution frameworks, EI-DDPM and EI-FM, to address it. The proposed methods transfer from existing works’ mapping individual samples to predictions into this work’s mapping an observation distribution to its corresponding posterior distribution. To our knowledge, this general framework for distribution-to-distribution mapping has not been previously defined or explored in the inverse problem context. We kindly request the reviewer to please reconsider the score and evaluation in the light of these clarifications.

---

### Official Review · Reviewer_i4bK · 2025-10-31

**Soundness:** 3
**Presentation:** 3
**Contribution:** 2
**Rating:** 4
**Confidence:** 4

**Summary:**

The authors study the ensemble inverse problem (EIP) which is described as follows. For training, we are given access to several collections $\mathcal{D}_m$ of (x, y) pairs, where the y are observations of a noisy forward process on the ground truth variable x. In each collection, the ground truth variables x are sampled from a different prior distribution $p_m$. At test time, we are given a new collection $\mathcal{Y}$ of only the observations y with x sampled from an unknown, different prior distribution $p$. The goal is to approximate either the prior distribution $p$ on x, or the posterior on x conditional on the observed y, i.e. $p(x|y)$.

The authors propose two algorithms for solving this problem. These algorithms can be viewed as a modification of the standard ODE and PDE diffusion model procedures, where one trains a neural network to predict the noise component of the target x + Gaussian noise, and progressively remove the noise over many steps to recover the original x. Their modification to this approach is to provide two additional inputs to the denoising network. The first is the observation y from the forward process corresponding to the x which is to be denoised. The second is an encoding of the *collection* of observations $\mathcal{Y}$ corresponding to the point to be denoised, to help give the denoising network some information about the prior.

The authors then test their algorithms on a synthetic Gaussian example, simulated particle physics data, and MNIST, and show that their method generally performs comparably or better than relevant baselines. In particular, it is often able to generalize to recover priors which were not seen during training.

**Strengths:**

For the most part, the paper is clearly written and easy to follow.

The experiments cover a diversity of settings. The synthetic Gaussian experiment is easy to visualize and provides some good intuition, and also shows explicitly how the method can generalize beyond priors not seen during training. The particle physics experiments are helpful as the authors use problems from this field to motivate EIP. The MNIST experiment shows that the proposed method may also have applicability in other realistic settings, with the added difficulty of high dimensionality. Finally, the numerical results are generally favorable for the proposed methods.

The intuition for the proposed method is clear and it should be easy to incorporate it into general diffusion model training pipelines.

**Weaknesses:**

The methodological novelty is somewhat limited. As described in the summary, the method amounts to providing two additional inputs to the denoising network for a standard diffusion model setup. While it is intuitive that the inclusion of these two quantities can encode information about the prior to be reconstructed in a form which is usable by the denoising network, the paper also does not attempt to place this intuition on more rigorous footing, e.g. with theoretical results or even informal mathematical derivations.

In the first line of the abstract, the authors claim to introduce a novel inverse problem in EIP. However, on lines 96-97, the authors acknowledge that EIP-II has been directly considered by Pazos et al. (2025). The methods the authors introduce are also developed specifically for EIP-II (line 174, beginning of the Methods section), which seems to contradict the claim in the abstract.

On the topic of related work, the work is closely connected to the extensive literature on using diffusion models for Bayesian inverse problems, and many methods from this literature could be used as baselines especially for the image denoising experiment. See Appendix A of the following paper for a recent extensive overview:

>Chen, Haoxuan, et al. "Solving inverse problems via diffusion-based priors: An approximation-free ensemble sampling approach." arXiv preprint arXiv:2506.03979 (2025).

Finally, while the experiments are generally favorable for the proposed methods, this is not always the case. For example, in Fig. 3, the authors claim that EI-FM has superior performance compared to all other realistic baselines across the full range of $\gamma$. However, it actually looks like both Omnifold-best and Omnifold-combine perform better in the $|\gamma|\geq 0.75$ (highest and lowest) ranges. Confidence values for Fig. 3 and the numerical results in Table 2 are also not provided.

**Questions:**

Are there any rigorous results which can be given on the ability of the proposed methods to recover the ground truth?

On line 323 (bottom of pg. 6), the cFM-$\gamma$ method is mentioned presumably as an "unrealistically good" baseline method, as it has direct access to the prior information. However, I didn't see a full description of this method. How exactly does this method work?

---

> ### Author Response · Authors · 2025-11-18
>
> 1. The main novelty of the proposed methods lies in how to use the information contained in the observation set to help construct the posterior and the extension from sample-wise posterior sampler into distributional-dependent posterior sampler.  As far as we know, this is the first work focusing on incorporating ensemble information  extracted by neural networks into the modeling of the posterior.
>
>     In traditional regression, a model generalizes because it learns a functional relationship that holds for all samples drawn from the same underlying distribution, rather than memorizing specific inputs. This work extends the same principle to the **distributional level**: instead of mapping individual samples to predictions, EI models map an observation distribution to its corresponding **posterior distribution** $p(x∣y,\mathcal{Y})$. Since the model learns a stable distribution-to-distribution relationship—analogous to learning a sample-to-sample function in regression—it naturally generalizes to unseen observation sets drawn from the forward operator and a prior family. We will add this intuition into the paper.
>
>     Rigorous theoretical proofs are indeed useful to develop bounds on the performance of the proposed methods and are promising future directions.
>
> 2. Pazos et al. (2025) mainly focuses on the unfolding task in HEP, which is an important real-world EIP problem. Specifically, it proposes to model the posterior and this fits the EIP-II setting. Although the unfolding task is an EIP problem, it does not mention the concept of general EIP problem. Therefore we say  in the paper (line 96):
>
>     “ **In the context of unfolding**, EIP-II setting has recently been considered directly in Pazos et al. (2025).”
>
> 3. Thanks for recommending this related work. However, this ensemble sampling approach cannot be applied to the setting in our work. In our work, the forward operator is assumed to be fixed and unknown, and only truth-observation pairs are provided. In the ensemble sampling work, the algorithms explicitly requires log-likelihood term $\mu_y(x) = -\log p(y|x)$. Without additional model assumptions, no finite sample of truth-observation pairs uniquely determines an analytic form of $p(y|x)$.
>
> 4. We agree with the reviewer that in the toy Gaussian experiment, EI-DDPM/FM behave slightly worse than Omnifold for extreme values $|\gamma|>0.9$. $\gamma$ reflects the covariance of the 2D Gaussian distribution and the distribution will be close to a straight line when $|\gamma|\rightarrow 1$. Therefore the prior when $|\gamma|\rightarrow 1$ deviates significantly from the seen priors during training, leading to slightly worse generalization performance.
>
>     In the unfolding task, the SWD is computed between all the available truth-level data and detector-level data / recovered data ($10^6$ samples for each physics process). Several approaches based on FM framework have deterministic sampling results. Therefore the confidence level of the unfolding task is not computed and reported.
>
>     We will add a figure to show the confidence level of Fig. 3 in the appendix.
>
> Questions:
>
> 1. It will be indeed useful to develop bounds on the performance of the proposed methods. On top of that it will be even more useful to come up with an algorithm independent bound (information theoretic) on the EIP problem and see if the proposed methods are able to achieve those bounds. Nevertheless, at present, this development is beyond the scope of this work.
>
> 2. We have the description  of cFM-$\gamma$ on line 332-333: “*Besides the mentioned baselines, we also evaluate cFM-$\gamma$, which is built based on cFM, but additionally conditioned on the latent information $\gamma$. cFM-$\gamma$ assumes direct knowledge of the priors.*” This shows that cFM-$\gamma$ conditions on both the observation and the direct prior information.

---

### Official Review · Reviewer_RwgK · 2025-11-03

**Soundness:** 2
**Presentation:** 2
**Contribution:** 1
**Rating:** 2
**Confidence:** 4

**Summary:**

The paper formalizes a new inverse problem termed the Ensemble Inverse Problem (EIP), where an additional set of observations is incorporated into the inversion process. The authors propose to address EIP using conditional generative models, specifically diffusion and flow-matching–based approaches (EI-DDPM and EI-FM). The model is evaluated on three tasks—synthetic 2D Gaussian problem, particle-physics data unfolding, and MNIST image inversion—and demonstrates superior performance.

**Strengths:**

The paper provides a clean formalization of the ensemble inverse problem, emphasizing inference across multiple priors with a shared but unknown forward operator.

**Weaknesses:**

- While the EIP formulation is interesting conceptually, its practical relevance to real-world inverse problems is not clear. The authors mention applications in high-energy physics and inverse imaging, but for readers unfamiliar with high-energy physics, the motivation in that domain is difficult to assess. For inverse imaging problems, it is unclear to me how the EIP problem setting arises in practice.
- Despite the new terminology, Algorithms 1 and 2 follow standard conditional diffusion training and sampling procedures. The only notable modification is the addition of a Set Transformer encoder for $\mathcal{Y}$, and even this encoder is used only in one experiment (the particle-physics data unfolding).
- The "high-dimensional" inverse imaging experiment only uses MNIST (28x28 dimensions), which is far too simple to demonstrate the proposed model’s utility for serious inverse imaging problems.
- The proposed method requires retraining for each new forward model, as it depends on paired truth–observation datasets. In many practical settings, generating such datasets or retraining large diffusion models may be infeasible.

**Questions:**

- Can authors provide more concrete examples of how EIP corresponds to a real imaging scenarios (e.g. MRI[1], deblurring[2], etc) as well as experimental evidence?
- Why do the authors tailor the algorithms specifically for DDPM and FM? It's known that diffusion model is equivalent to FM up to a simple reparameterization for Gaussian prior setting [3]. Are there specific empirical motivations for retaining both?

[1]: Sriram, Anuroop, et al. "End-to-end variational networks for accelerated MRI reconstruction." _International conference on medical image computing and computer-assisted intervention_. Cham: Springer International Publishing, 2020.

[2]: Mardani, Morteza, et al. "A variational perspective on solving inverse problems with diffusion models." _arXiv preprint arXiv:2305.04391_ (2023).

[3]: [Diffusion Models and Gaussian Flow Matching: Two Sides of the Same Coin](https://d2jud02ci9yv69.cloudfront.net/2025-04-28-diffusion-flow-173/blog/diffusion-flow/)

---

> ### Author Response · Authors · 2025-11-18
>
> 1. While the high-energy physics (HEP) unfolding task may be unfamiliar to some readers, it is a major real-world problem that drove this research. See also the comments from the first reviewer.
>
>     Unfolding is a widely studied practical problem in HEP, with significant attention in both methodology and applications, as shown in 1911.09107, 2406.01507 and 2308.1235. The image experiment on MNIST digits serves as an intuitive example to show that EI-DDPM/FM can be applied to image domains.
>
>     *To further illustrate the real-world applications, we plan to add a real-world audio inverse problem and update the results by Dec 03.*
>
> 2. The main novelty of the proposed methods lies in how to use the information contained in the observation set to help construct the posterior and the extension from sample-wise posterior sampler into distributional-dependent posterior sampler.
>
>    **We  disagree with the reviewer’s comment** that the encoder for ensemble information is used in only the unfolding experiment. In all three experiments, EI-DDPM/FM corresponds to Alg. 1, which requires a permutation invariant encoder for ensemble information.
>
> 3. We agree with the reviewer that the MNIST experiment does not reach the complexity of real-world images. For real world applications, we plan to add a real-world audio inverse problem and update the results by Dec 03.
>
> 4. We agree that the proposed approaches require retraining for new forward operators and require truth-observation pairs. However, the setting of fixed unknown forward forward operators and available truth-observation pairs is a common setting, as shown in 2406.01507 and 2308.1235. Other settings requiring explicit forward operators are actually stricter compared with the settings in this paper since one can create truth-observation pairs via the forward operator. **Therefore assuming the availability of truth-observation pairs is not impractical.**
>
>     We refer the reviewer to Table 1, in which all references assume that the dataset is available or the forward operator is available - There are indeed approaches that work for different forward operators, such as  1911.09107 and 2402.07808. However, they are iterative approaches that require repeatedly adjusting the simulated events to match data likelihood, till convergence. This process is required even for the same forward operator, causing longer inference time compared to this work. Therefore there is a trade-off between these two  approaches – If the forward operator remains identical, this work can be a good choice due to shorter inference time. However, if the forward operator changes frequently (but known), iterative approaches are better since they do not require pre-training large diffusion models.
>
> Questions:
>
> 1. The application of this work mainly has the following two  features,
>
>     (1) The priors come from a mixture of distributions and the forward operator is fixed.
>
>     (2) A single observation cannot carry the complete ensemble information, i.e., from a single observation, one is unable to determine the prior. Otherwise, ensemble information from the set is unnecessary.
>
>     Yes, at present, other than the HEP data set, we have not formulated another real-world case and specifically a real-world inverse imaging case that fits this setting naturally, instead, we plan to add a real-world audio inverse problem and update the results by Dec 03.
>
> 2. We agree with the reviewer that if either one of DDPM and FM works by incorporating the ensemble information, it is generally expected that the other also works. However, the implementation of  two algorithms differ and we tailor the algorithms for both DDPM and FM to show that this approach is compatible for these two popular frameworks. The empirical results display close performance of the two methods as shown in the unfolding and MNIST experiment.

---

### Official Review · Reviewer_Ay8h · 2025-11-04

**Soundness:** 3
**Presentation:** 2
**Contribution:** 3
**Rating:** 6
**Confidence:** 3

**Summary:**

This paper introduces a formulation for inverse problems, which the authors term the "Ensemble Inverse Problem" (EIP). The goal of EIP is to recover a "truth" distribution, $p(x)$, given a set of observations, $\mathcal{Y}$, which are generated by feeding samples from $p(x)$ through a fixed but unknown forward model and adding noise to them to build a likelihood distribution $p(y|x)$. This problem is common in science (e.g., "unfolding" in particle physics) and imaging, especially when the prior $p(x)$ is unknown.
The authors distinguish two versions of the problem:
1. EIP-I (Prior): Recover the prior $p(x)$ from the ensemble $\mathcal{Y}$.
2. EIP-II (Posterior): Recover the posterior $p(x|y)$ for a single observation $y$, using the entire ensemble $\mathcal{Y}$ as context.

The core methodological contribution is what the authors call "ensemble inverse generative models," that are trained to approximate $p(x|y, \mathcal{Y})$. This is achieved by conditioning a generative model (e.g., DDPM or Flow Matching) on both the single measurement $y$ and a learned, permutation-invariant embedding of the entire observation set $\mathcal{Y}$, $\phi_w(\mathcal{Y})$. To enable generalization, the model is trained on a collection of datasets from different priors, all passed through the same unknown forward operator.
The authors demonstrate through experiments on synthetic data, a 7D particle physics unfolding problem, and a synthetic image inversion task that their method achieves state-of-the-art results. They show it can successfully generalize to unseen priors that were not included in the training data, a task where naive conditional models fail.

**Strengths:**

1. The formalization of the "Ensemble Inverse Problem" (EIP) is a significant contribution. This is essentially a new "in-context learning" or "meta-learning" framework for scientific inverse problems, which is potentially impactful. The proposed solution—conditioning on a learned, permutation-invariant embedding of the observation set $\mathcal{Y}$—is an elegant and general way to solve the problem. Instead of using hard-coded statistics (like moments), it allows the model to learn the most relevant features of the observation set that define the prior, and does not require knowledge of the forward model.
2. The experiments are designed to test the paper's central claim.
    * In the synthetic 2D Gaussian and MNIST-mixture experiments, the model is explicitly trained on a disjoint set of priors and tested on the "in-between" region (e.g.,  interpolation between seen regions).
    * The results in Figures 3 and 6 are unambiguous . The proposed method (EI-FM/EI-DDPM) successfully generalizes and solves the problem in the unseen regions, while the baseline (cFM/cDDPM) that lacks the ensemble information completely fails. This is a very powerful demonstration.
3. The method achieves SOTA results on a real world HEP unfolding task, outperforming multiple baselines (including the similar GDDPM) on 4 unseen physics processes. This shows the approach scales and is practical for complex scientific data.

**Weaknesses:**

1. Ambiguous Experimental Setups: A few points in the paper would need to be clarified, specifically about the experimental setups.
    - For the HEP experiment:  My understanding was that the paper's premise is that the forward model and the noise model that allow building $p(y|x)$ need to be fixed. However, the data description mentions using "various parton distribution functions and parton shower models". This sounds like the underlying physics, and thus $p(y|x)$, is changing across the different datasets in M. This would really need clarification.

    - For the MNIST digit mixture problem, it’s not clear what is the goal of the task. It is to recover one of the deblended digits (or both), or is it to recover the blended mixture at some given time (making it much more like an inpainting+denoising task)? From the text I understood it was the former, but Figure 6 seems to suggest the latter. This needs to be clarified in the text.
2. Limited baseline comparisons: The paper positions itself as a general solution for inference, but I was surprised to see that baseline comparisons were limited. While it’s true that most inference methods that have been developed with scientific applications in mind rely on knowledge of a forward model (except on naive conditioning on observation with diffusion and flow matching), with the paired dataset that is assumed to be available, it would be natural to train an emulator for the forward model, which could then be used in a standard inference framework like SBI. This is the natural baseline to compare the proposed method against (and, as I understand, a standard method for HEP applications as well). This would potentially be impactful since SBI methods are known to struggle with covariate shifts, so it’d be natural to show the strength on the proposed method against this specific baseline, and not including it is a clear lack in the paper in its current state (7dimension is very low-dimensional, definitely within the applicability of SBI methods). Moreover, once an emulator has been trained, it would be natural to compare against population-level inference frameworks to empirically adapt or learn to correct misspecified priors, such as 2402.07808, or more general expectation-maximization methods like those proposed in 2405.13712 or 2407.17667. Seeing how the proposed methods compare to these more recent proposed approaches is a natural question for a practitioner.

3. Limited Evaluation Metrics: The paper evaluates the quality of their inference using metrics like SWD, WD, MSE, and SSIM. These metrics compare bulk properties, means, or medians, and these can miss key differences between high-dimensional distributions.  Coverage tests like TARP (2302.03026) or accuracy scores like PQMass (2402.04355) would be needed to assess if the inferred posteriors/priors here accurate in terms of the entire distribution. This is especially a problem for the image-space application with MNIST, since just the MSE or the SSIM of the mean of the posterior is assessed. The mean, specifically in image space, could look nothing like any individual sample and this could affect the results specifically for structural integrity. It could also favor methods that are overconfident.

4. Unclear Scaling: The paper claims scalability, but the "high-dim" MNIST test is only $28 \times 28$. A real study of how this method would scale in real high-dimensional settings (e.g., ImageNet-scale), is missing.

5. Limited scientific applications: The performance of the method is demonstrated in a single scientific context (HEP), but the generals claims are that the method is useful for general scientific applications. Applications to other scientific contexts like those in the benchmark in 2503.11043 would really strengthen the paper. Other domains of applications where a forward model is generally not known are, for example, weather forecasting, climate modelling, and cosmology.

**Questions:**

1. About the fact that the physical model seems to be required to be fixed across M: In most scientific applications I’ve seen, when the true physical model is unknown, what is available is typically a whole menu of physical simulators that approximate this underlying physical model to various degrees of fidelity (and various levels of compute cost). It’s those approximate simulators that allows building the mapping from x to y. But in that context, 1- it’s never guaranteed that the true mapping in within the set of simulations we can access, 2- the physical model would be changing between simulations and real experiment where we want to apply the trained model. This is related to point 1 in the weaknesses, can you clarify the assumptions you are working under, and if the physical model does need to be fixed, I’d like it to be explained more clearly how this is supposed to applied to a realistic real-world experimental setup.

2. Similarly, it’s not clear from the paper if the proposed method would work is the different priors were over a heterogenous parameter space, meaning that $p_m(x)$ have different dimensionalities for $x$ across different $m$? This is another context where I could see this applied in a scientific context where one would like to do Bayesian model comparison/fit different model with varying number of parameters. Is that something that could be possible?
3.  I was surprised to see that GDDPM-v (with more information) performs worse than GDDPM (with less). This suggests potential overfitting or non-optimal training, which weakens the claim of SOTA performance. Adding information should not impair recovery if the model is well-trained, do you have an explanation for why this is not the case?
4. Could you provide more details on how the experiments were set up? For example, for Fig 2, how was the prior obtained from observations for the conditional diffusion model and flow matching? Was it by getting the MAP or the mean of the posterior samples for multiple observations? Also, could you provide more details on how the conditioning was done for the other models for the MNIST experiment?

---

> ### Author Response · Authors · 2025-11-18
>
> 1. Ambiguous Experimental Setups
>
>     (1) The reviewer’s understanding that $p(y|x)$ is fixed is correct. The descriptions "various parton distribution functions and parton shower models" are for the priors, not the forward operators. To clarify this, we will make the following revision :
>
>       “*The training data consists of pairs of truth-level and detector-level jet vector pairs. The truth-level jet vectors come from* $18$ *different physics processes, including various parton distribution functions and parton shower models, and the detector effects are identical across all truth-level jet vectors*. “
>
>     (2) The objective of the MNIST digit mixture problem is:
>
>     “*At the inference stage, given a set of blurred images, which come from a common prior at an unknown interpolation  time* $t$, *the objective is to recover the corresponding clean images for each blurred image in the set*.”
>
>     We will add this sentence to clarify the goal in the paper.
> 2. Limited baseline comparisons
>
>     In the paper, we assume that, at inference time, the forward operator is unavailable, therefore, we did not include baselines that require a known forward operator.
>
>     However, we do agree with the reviewer that one important baseline is that one first train a simulator for the forward operator from the provided truth-observation pairs, then apply approaches that use the forward operator, such as the maximum-entropy estimation method in 2402.07808 and the expectation-maximization method in 2405.13712.
>
>    *We plan to add a baseline with a simulator of the forward operator coupled with the approach in 2402.07808 for the physics unfolding experiment by Dec 03, and we will update the results here.*
>
>     **We want to bring to the reviewers’ attention that  a  key difference between EI-DDPM/FM and the approaches requiring forward operators is that EI-DDPM/FM is a non-iterative approach and forward operator-based approaches (including Omnifold) are usually iterative since they require repeatedly adjusting simulated events to match data until convergence. An advantage of non-iterative approaches over iterative approaches is significantly reduced inference time.**
>
> 3. Limited Evaluation Metrics
>
>     Thank you very much  for recommending two important metrics TARP and PQMass. Using paired simulation samples as the reference distribution makes TARP metric a valid and sufficient test of posterior correctness. Computing PQMass between posterior samples from the generative model and reference posterior samples provides a rigorous test of posterior correctness.
>
>     *We will add the TARP metric for all experiments before Dec 03, and we will update the results.*
>
> 4. Unclear Scaling
>
>     We agree with the reviewer that MNIST images are not as “high-dimensional” as real-world images. We chose MNIST because the priors of digit mixtures can be easily constructed, and the resulting mixed digits remain visually distinguishable and have clear digit structures. In contrast, images of greater complexity from two categories tend to become visually “blurred” and do not display meaningful patterns when mixed.
>
>     We leave for future work the search for a high-dimensional inverse imaging task , which naturally fits in the EIP settings, having sufficient priors and unknown but fixed forward operators.
>
> 5. Limited scientific applications
>
>     Thank you  for recommending 2503.11043, which is a powerful benchmark for inverse problems. EIP settings are necessary when the task has following features
>
>     (1) The priors come from a distribution over  (prior) distributions and the forward operator is fixed.
>
>     (2) A single observation cannot carry the complete ensemble information, i.e., from a single observation, one is unable to determine the prior.
>
>     2503.11043 does not contain a task that has these features. However, we looked for another application, viz., an audio inverse problem, that can illustrate the effectiveness of EI-DDPM/FM that we plan to take up.

---

> > ### Author Response · Authors · 2025-11-18
> > **Response for Questions**
> >
> > Questions:
> >
> > 1. The  aim of this paper was to mainly formalize the EIP and to begin with, mainly focus on the setting, in which there is only a single forward operator that is fixed but unknown. The learning is through truth-observation pairs resulting  from different priors. At the inference time, a set of observations, coming from a single unknown prior is provided, and the objective is to recover for each observation in the set. As stated above, we were guided by the need for a faster inference time without the need to retrain anything.
> >
> >     If the setting changes, namely,  there are  several possible forward operators or a parameterized family of forward operators, we believe our methods can be extended to handle this situation but will likely require some structural assumptions on the family of unknown priors and on the family of forward operators.
> >
> >     In other words, this is related to finding a (unique) factorization of the marginal into a prior and a forward operator, which is only possible under some structural assumptions. See e.g. works related to non-negative matrix factorization to recover topics or to recover spectral signatures in hyperspectral imaging. We believe that exploring this situation is indeed a promising direction but is currently beyond the scope of the present work.
> >
> > 2. The proposed method only requires minor modification before it can be applied to cases with priors of different dimensions. First, pad all data of different dimensions to their highest dimension. Second, use the same structure to train EI-DDPM/FM models. Third, design a projection rule, to project the data back into their original dimension (value at extra dimensions will be close to 0). We agree with the reviewers that this can be potentially related to Bayesian model comparison and we appreciate the pointer.
> >
> > 3. In the unfolding case, the distribution of the 7-dim data $x$ is determined by the distribution of its first component $x_1$. However, provided with the distribution of $x_1$, one is unable to recover the distribution of the 7-dim data. This is some extra knowledge of the prior. As we mentioned in the paper “GDDPM implicitly assumes that $p_T$ contains the complete distributional information of the 7-component vector.” GDDPM-v accepts more, but unnecessary information. This increases the dimensions of the conditional information and may cause overfitting to these redundant features. A “well-trained” model can possibly exist since it has all “useful” information, however, the training objective of minimizing the empirical loss can cause overfitting and make it hard to converge to a “well-trained” point.
> >
> > 4. At inference time, we are provided with a set of observations and we recover for each observation via Alg. 2. So the prior from diffusion models shown in Fig. 2 is the collection of $40000$ recovered observations. The other models for MNIST experiments (cFM, cDDPM, SBUnfold) are existing works without considering the information from the observation set, i.e., the ensemble information. Therefore, their conditional information only includes the blurred image.

---

### Author Response · Authors · 2025-12-03

We thank the reviewers for their detailed comments and feedback.

As a general response, we first want to clarify the novelty and contribution of this work.

1. Our contribution lies in the formalization of the Ensemble Inverse Problems (EIP) – a category of inverse problems, in which the prior is unknown, but an observation set is accessible. EIP manifests itself in several recent domains but was previously not formally defined and is currently an insufficiently explored problem.

2. After formalizing EIP and bringing out its practical and theoretical relevance,  we provide two methods EI-DDPM and EI-FM, to address it. The proposed methods exploit the conditional generative models BUT with a novel feature that takes ensemble information into account. While simple, we show they are effective.

3. Although some reviewers may not be familiar with the EIP problem that arises in high-energy physics, we want to emphasize that it is a major real-world problem [1,2,3] that drives this research.

**We did anticipate the questions regarding the inverse problems literature, but we want to emphasize  that EIP is fundamentally different compared with the traditional set-up, and our emphasis is on non-iterative methods that do not explicitly use the forward operator at inference time.**

New evaluation:

Based on the suggestion of reviewers Ay8h and 2G23, we added

1.  **A new baseline Sourcerer [4] in Sec. 3.2 and 3.3**, in which a surrogate for the forward model is trained based on the truth-observation pairs, and it solves inverse problems that jointly maximizes entropy and minimizes sample-based distance.

2. Based on the suggestion of Ay8h, we added another metric, i.e., TARP coverage [5] for evaluating posterior samples in Sec. B.5.

While we fully understand that discussion with the reviewers is not permitted at this stage, we believe that we have addressed most of the reviewers’ comments and suggestions in our responses. We kindly invite the Area Chair to carefully consider the comments from all reviewers in assessing whether concerns have been raised based on the revised version of the manuscript (changed parts are marked in blue), and to make a balanced judgment based on our work’s overall merit.

[1] Anders Andreassen et al. “Omni-fold: A method to simultaneously unfold all observables”.

[2] Sascha Diefenbacher  et al. “Improving generative model-based unfolding with schr¨odinger bridges”.

[3] Camila Pazos et al. “Towards universal unfolding of detector effects in high-energy physics using denoising diffusion probabilistic models”.

[4] Julius Vetter et al. “Sourcerer: Sample-based Maximum Entropy Source Distribution Estimation”.

[5] Pablo Lemos et al.”Sampling-based accuracy testing of posterior estimators for general inference”.

---

### Meta-Review · Area_Chair_VS6w · 2026-01-06

**Summary:**

The paper formalizes the Ensemble Inverse Problem (EIP), where an additional set of observations is incorporated into inversion, and proposes conditional generative solutions via diffusion and flow-matching (EI-DDPM, EI-FM).

While the formulation is conceptually interesting, the reviews are mostly negative, mainly due to unclear practical relevance and limited methodological novelty. The claimed applications (high-energy physics, inverse imaging) are not sufficiently motivated for a broad ICLR audience, and it is not clear how the EIP setting commonly arises in real inverse imaging. Methodologically, Algorithms 1–2 largely follow standard conditional diffusion training/sampling; the main extra component is a Set Transformer encoder for the observation set, which is only used in one experiment. The experimental evidence is also not strong enough: the “high-dimensional” imaging result is only on MNIST (28×28), and even the sole positive reviewer notes ambiguous setups, limited baselines and metrics, and limited scientific validation. Overall, the current submission falls short of ICLR standards.

**Reviewer Concerns:**

.

**Reviewer Scores:**

.

---

### Decision · Program_Chairs · 2026-01-26

Reject